# D2C-HRHR: Discrete Actions with Double Distributional Critics for High-Risk-High-Return Tasks

## Abstract

Tasks involving high-risk–high-return (HRHR) actions, such as obstacle crossing, often exhibit multimodal action distributions and stochastic returns. Most reinforcement learning (RL) methods assume unimodal Gaussian policies and rely on scalar-valued critics, which limits their effectiveness in HRHR settings. We formally define HRHR tasks and theoretically show that Gaussian policies cannot guarantee convergence to the optimal solution. To address this, we propose a reinforcement learning framework that (i) discretizes continuous action spaces to approximate multimodal distributions, (ii) employs entropy-regularized exploration to improve coverage of risky but rewarding actions, and (iii) introduces a dual-critic architecture for more accurate discrete value distribution estimation. The framework scales to high-dimensional action spaces, supporting complex control domains. Experiments on locomotion and manipulation benchmarks with high risks of failure demonstrate that our method outperforms baselines, underscoring the importance of explicitly modeling multimodality and risk in RL.

## 1 Introduction

Reinforcement Learning (RL) typically uses discrete action spaces for discrete tasks and continuous spaces for continuous tasks. For discrete tasks, such as Atari games, Q-learning evaluates only a small number of actions. For continuous tasks, such as robotic motion control, evaluating all actions is infeasible. Discrete-action models can suffer from the *curse of dimensionality* (Kober et al., 2014; Lillicrap et al., 2016), while continuous-action models output actions directly (Vaserstein, 2014; Lillicrap et al., 2016; Schulman et al., 2017; Fujimoto et al., 2018; Haarnoja et al., 2018; Kuznetsov et al., 2020).

Many real-world RL tasks involve high-risk-high-return (HRHR) scenarios (Fig 1 (a) top right), where the highest rewards occur only in risky regions of the action space (Fig 1 (c) region $\Omega_1$). Examples include parkour locomotion, obstacle crossing, or contact-rich robotic manipulation. Standard RL methods often assume unimodal Gaussian policies and scalar critics, which bias learning toward safer actions (Fig 1 (c) region $\Omega_2$) and fail to capture high-reward regions in HRHR tasks. We formally define HRHR tasks and show that Gaussian policies cannot guarantee convergence to the optimal solution.

Recent work revisits discrete actions for continuous tasks. Andrychowicz et al. (2020) and Tang & Agrawal (2020a) demonstrated improved on-policy RL via discretization, and Seyde et al. (2021) achieved state-of-the-art results using extreme discretization akin to Bang-Bang control. These successes highlight discrete action spaces' potential for multimodal exploration and capturing high-reward actions in HRHR scenarios.

Based on these observations, we propose a discrete-action framework (i) discretizes continuous action spaces to approximate multimodal distributions (Fig 1 (d)), (ii) employs entropy-regularized exploration to improve coverage of risky but rewarding actions, and (iii) introduces a dual-critic architecture for more accurate discrete value distribution estimation a discrete actor with twin discrete critics ((Fig 1 (e)). With them, our model can capture actions in localized high-reward regions surrounded by low-return or even harmful outcomes (Fig 1 (c) region $\Omega_1$).

Empirical results on baseline benchmarks, including locomotion tasks with a high risk of falling and manipulation tasks with a high risk of failure, suggest that that our method outperforms baselines in HRHR scenarios.

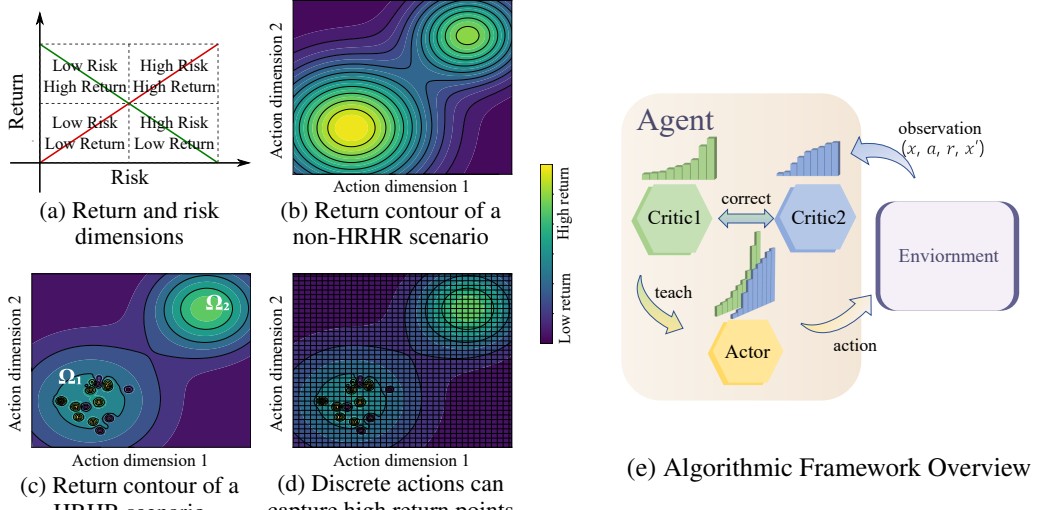

(a) Return and risk dimensions

(b) Return contour of a non-HRHR scenario

(c) Return contour of a HRHR scenario

(d) Discrete actions can capture high return points

(e) Algorithmic Framework Overview

Figure 1: Return and risk are in two dimensions. In an ideal task, higher expected returns are associated with lower-risk actions, as shown by the green line in (a) and panel (b). However, in tasks where favorable outcomes are intertwined with risk, the highest returns have to be extracted among high-risk actions, as shown by the red line in (a) and panel (c). Actions sampled from Gaussian distribution can be hard to capture the high return points during learning as the expectation on the distribution is low, but discrete actions can capture the actions.

## 2 RELATED WORK

Distributional RL models focus on modeling the distribution of cumulative rewards rather than only an expected scalar. C51 (Bellemare et al., 2017) employs a discrete value distribution for building its critic network. QR-DQN (Dabney et al., 2018b) and IQN (Dabney et al., 2018a) utilize quantile regression to detail the distribution of stochastic reward returns.

Some previous works resemble a few aspects of our work, but differently. D4PG (Barth-Maron et al., 2018) proposed a C51 with an actor module. Our work, however, diverges by investigating discretized action spaces, moving away from the conventional assumption that action variables conform to a normal distribution. SAC-Discrete (Christodoulou, 2019) broadens the scope of SAC to discrete action spaces, thus enhancing the model's capacity to use action entropy for exploration.

There are a few more works using discrete actions for continuous tasks. Neunert et al. (2019) explores a feasible approach for the unified control of discrete and continuous action variables based on the MPO algorithm. Tang & Agrawal (2020b) advocate for the discretization of continuous action spaces, which can enhance the performance of on-policy algorithms such as PPO. Luo et al. (2023) emphasize the benefits of discretizing action spaces in offline reinforcement learning and examine potential solutions. Metz et al. (2019) decompose multi-dimensional action variables into a sequence of decision-making processes for discrete variables. Farebrother et al. (2024) argues that the cross-entropy loss function, compared to the mean squared error loss function, is more effective for training critic networks in reinforcement learning. None of these works use double distributional critics. In our experiments, we used some of these algorithms for comparison.

## 3 MODELS & METHODS

In this section, the high-risk-high-return (HRHR) scenario in RL tasks is formally defined, and why a Gaussian policy cannot guarantee an optimal result is proved. According to the theoretical analysis,

we proposed our model which (i) discretizes continuous action spaces to approximate multimodal distributions, (ii) employs entropy-regularized exploration to improve coverage of risky but rewarding actions, and (iii) introduces a dual-critic architecture for more accurate discrete value distribution estimation. For consistency, the mathematical notation in this paper can also be referenced to the Table of Mathematical Symbols in the Appendix 6.1.

### 3.1 HRHR SCENARIO

Here we define the HRHR scenario in RL tasks as, given a state of the environment, a region of action space contains a set of high return actions and a set of low return actions, which results in that, the average return of this region is lower than that of another region whose highest return is lower than the former's highest return. In Figure 1 (b) and (c), we present the contour plots of HRHR scenarios and non-HRHR scenarios.

Consider a reinforcement learning agent operating in an environment with state space $\mathcal{S}$ and action space $\mathcal{A}$. Let $Q : \mathcal{S} \times \mathcal{A} \to \mathbb{R}$ denote the action-value function, where $Q(s, a)$ represents the expected return when taking action $a$ in state $s$.

**Definition 1** (High-Risk-High-Return Scenario). *For a given state $s \in \mathcal{S}$, we say there exists a high-risk-high-return (HRHR) scenario if there exist measurable regions $\Omega_1 \subseteq \mathcal{A}$ and $\Omega_2 \subseteq \mathcal{A}$ with positive measure ($\mu(\Omega_1) > 0$, $\mu(\Omega_2) > 0$) satisfying the following conditions:*

$$\sup_{a \in \Omega_1} Q(s, a) > \sup_{a \in \Omega_2} Q(s, a) \qquad \mathbb{E}_{a \sim \mathcal{U}(\Omega_1)}[Q(s, a)] < \mathbb{E}_{a \sim \mathcal{U}(\Omega_2)}[Q(s, a)] \tag{1}$$

*where $\mathcal{U}(\Omega_i)$ denotes the uniform distribution over region $\Omega_i$, and $\Omega_1$ is called the HRHR region. If $\mathcal{A}$ is continuous, the expectations are computed as:*

$$\mathbb{E}_{a \sim \mathcal{U}(\Omega_i)}[Q(s, a)] = \frac{1}{\mu(\Omega_i)} \int_{\Omega_i} Q(s, a) da \tag{2}$$

In this scenario, an RL algorithm should adjust the distribution of the action to let the expectation of the return measured on the action distribution be higher.

However, given a policy with the Gaussian actions, if the variance of the action is larger than the grain with the high rewards in the HRHR region, an RL algorithm could lead the policy to prefer $\Omega_2$ instead of $\Omega_1$.

**Definition 2** (High-Reward Grain). *A high-reward grain $\mathcal{G} \subseteq R_1$ is a connected component satisfying:*

$$\inf_{a \in \mathcal{G}} Q(s, a) > \sup_{a \in R_1 \setminus \mathcal{G}} Q(s, a) \quad and \quad \mathrm{diam}(\mathcal{G}) \leq \delta \tag{3}$$

*with $\delta > 0$ being the maximum grain diameter. The set of all high-reward grains in $R_1$ is denoted $\mathbb{G}_1$.*

**Theorem 1** (Policy Preference in HRHR Scenarios). *For Gaussian policy $\pi_\theta(\cdot|s) = \mathcal{N}(\mu_\theta(s), \sigma_\theta^2(s)I)$ in an HRHR scenario at state $s$ with high-reward grains $\mathbb{G}_1$ of diameter $\delta$, if $\sigma_\theta(s) > \delta$, then the gradient update satisfies:*

$$\langle \nabla_\theta J(\theta), \Delta\theta_{R_1} \rangle < \langle \nabla_\theta J(\theta), \Delta\theta_{R_2} \rangle \tag{4}$$

*where $\Delta\theta_{R_i}$ is the update direction toward region $R_i$, and $J(\theta)$ is the expected return. This implies gradient updates prefer $R_2$ over $R_1$.*

Thus, a policy with Gaussian actions can perform poorly in HRHR scenarios. In **Section 6.3.1** of the Appendix, we provide a detailed mathematical derivation to prove this point. In **6.3.2**, we will present different algorithms (SAC, TD3, C51) and our algorithm's process of predicting Q values in the form of schematic diagrams. This is closely related to the performance of the algorithm in handling HRHR scenarios. Additionally, in **Section 6.3.3** of the Appendix, we design an experiment called the "Trap Cheese Problem" to demonstrate the difference of decision between Gaussian policies and discrete policies in HRHR scenarios.

To address the challenges inherent in the HRHR scenarios defined above, we extended the basic distributed reinforcement learning algorithm, proposed the D2C-HRHR (Figure 1 (e)). For fundamentals of distributional reinforcement learning, please refer to 6.2.1 in the Appendix.

## 3.2 Multidimensional Discrete Actors

Our model employs a discrete action space across multiple dimensions. Instead of learning a single expected value of Q value, we learn the complete probability distribution, divide the possible reward range into a series of atoms, and then predict the probability of the reward distribution corresponding to each action on these atoms. In HRHR scenarios with high expected reward variance, this can more accurately identify the peak of expected returns.

A one-dimensional continuous action space is discretized into $m$ discrete action atoms $\{a_1, a_2, \cdots, a_m\}, m \in \mathbb{N}$, where $\mathbb{N}$ denotes the set of natural numbers. Then the discretization is applied to each dimension of a $n$-dimensional continuous action space, so an action in this new discrete space $\mathcal{A}$ can be noted as a matrix:

$$A \stackrel{D}{=} [\boldsymbol{a}_1, \boldsymbol{a}_2, \cdots, \boldsymbol{a}_n]^\mathsf{T} \tag{5}$$

where each row is one-hot coding of the corresponding action dimension. This shape is convenient to match the action probability distribution $\hat{A}$, which will be used as the output of policy network, and the sum of each row of $\hat{A}$ is 1. When we sample $A$ from $\hat{A}$, each row in $A$ is sampled from the probability of the corresponding row in $\hat{A}$.

In this action space, there exist $m^n$ discrete potential actions. Given such an extensive search space, employing exhaustive search methods such as those used by traditional DQN algorithms to find the maximal Q-value is not feasible. In this study, we propose modeling the agent's stochastic behavior within the action space $\mathcal{A}$ by utilizing an action probability matrix, therefore we set the actor as $\pi : \mathcal{X} \to \mathbb{R}^{n \times m}$.

$$\pi(\boldsymbol{x}) \stackrel{D}{=} \begin{bmatrix} p_{11}(\boldsymbol{x}) & p_{12}(\boldsymbol{x}) & \cdots & p_{1m}(\boldsymbol{x}) \\ p_{21}(\boldsymbol{x}) & p_{22}(\boldsymbol{x}) & \cdots & p_{2m}(\boldsymbol{x}) \\ \vdots & \vdots & \ddots & \vdots \\ p_{n1}(\boldsymbol{x}) & p_{n2}(\boldsymbol{x}) & \cdots & p_{nm}(\boldsymbol{x}) \end{bmatrix} \tag{6}$$

where $p_{ij}(\boldsymbol{x}) \geq 0$, $\sum_{j=1}^{m} p_{ij}(\boldsymbol{x}) = 1$ for $i = 1, 2, \cdots, n$, and $\boldsymbol{x}$ is an observed state. This $\pi$ characterizes a stochastic multi-dimensional discrete actor. A later section will detail how to use a neural network to approximate $\pi$. Please note that, in the original continuous action space, the action dimensions are independent, so in $A$, the elements between rows are also independent.

## 3.3 Clipped Double Q-learning for Discrete Value Distribution

Although using discrete actors can identify multiple expected peaks in HRHR scenarios. However, for distributed reinforcement learning algorithms with single criticism networks, such as C51, there are still issues in the HRHR scenario. Once they find areas with high expected returns (van Hasselt et al., 2015) , they will continue to learn in this direction. However, single criticism networks often overestimate the Q-value.

In this chapter, we will propose a novel dual value network suitable for discrete values. It can prevent the critic from overestimating the value of a high-risk action based on a few successful samples, thereby converging to a suboptimal strategy. By constructing two critic networks to estimate the discrete value distribution respectively and performing truncation operations during the update process, we can greatly improve the accuracy of the value network in evaluating action values. Although double critic networks have been used in some reinforcement learning methods, no one has applied them to distributed reinforcement learning before D2C-HRHR.

Double Q-learning for discrete distributional Q uses two critic networks, $\Theta_{\psi_1}(\boldsymbol{x}, \hat{A})$ and $\Theta_{\psi_2}(\boldsymbol{x}, \hat{A})$, and one actor network $\pi_\phi(\boldsymbol{x})$, where $\hat{A}$ is an action distribution matrix of a multidimensional discrete action space. $\boldsymbol{x}$ is the current state of environment.

It also has target networks $\Theta_{\psi_1'}(\boldsymbol{x}, \hat{A})$, $\Theta_{\psi_2'}(\boldsymbol{x}, \hat{A})$, and $\pi_{\phi'}(\boldsymbol{x})$ correspond to the main networks for stability in training. The subscripts above, $\psi_1$, $\psi_2$, $\phi$, $\psi_1'$, $\psi_2'$, and $\phi'$ denote the parameters of corresponding networks. Given a transition tuple $\boldsymbol{t} = (\boldsymbol{x}, A, r, \boldsymbol{x}')$, $r$ means reward from environment, $\boldsymbol{x}'$ is state of next observation. We consider how to effectively utilize these target networks to yield an updated estimation of the value distribution, $\Phi \hat{\mathcal{T}} \tilde{Z}(\boldsymbol{x}, \hat{A} | \Theta_{\psi_1'}, \Theta_{\psi_2'})$. With $\Theta_{\psi_1'}$ and $\Theta_{\psi_2'}$,

for $\hat{A}' = \pi_{\phi'}(\boldsymbol{x}')$, we derive $\Phi Z(\boldsymbol{x}', \hat{A}'|\Theta_{\psi_i'})$ as:

$$P(Z(\boldsymbol{x}', \hat{A}'|\Theta_{\psi_i'}) = z_k) \stackrel{D}{=} \frac{e^{(\Theta_{\psi_i'}(\boldsymbol{x}, \hat{A}'))_k}}{\sum_j^N e^{(\Theta_{\psi_i'}(\boldsymbol{x}, \hat{A}'))_j}} \tag{7}$$

A set of procedures is proposed to leverage the twin critic networks with a discrete value distribution, as shown in Figure 2. These procedures are presented in the Appendix 6.2.2 in the form of an algorithm table.

1. Firstly, the two critic networks estimate discrete value distributions according to $\boldsymbol{x}'$, respectively.
2. Secondly, the distributions are accumulated respectively.
3. Then for each category across the cumulative distributions, the one with higher probability is selected to form a new cumulative distribution.
4. Finally, each category of the new cumulative distribution, except the first one, is subtracted by the former one, mapping it back to discrete value distribution.

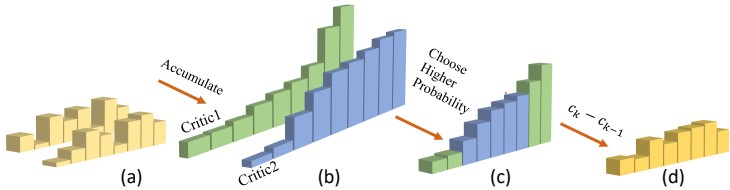

Figure 2: Clipped Double Discrete Value Distribution

The first and second procedures use the concept of "cumulative distribution". For a discrete value distribution $Z$, it can be depicted as follows:

$$P(Z \in \{z_1, z_2, \cdots, z_k\}) = \sum_{j=1}^k P(Z = z_j) \tag{8}$$

For the $k^{\text{th}}$ value atom, the third and fourth procedures can be presented by equation:

$$c_k \stackrel{D}{=} \max_{i=1,2} P(Z(\boldsymbol{x}', \hat{A}'|\Theta_{\psi_i'}) \in \{z_1, z_2, \cdots, z_k\})$$

$$P(\tilde{Z}(x', \hat{A}'|\Theta_{\psi_1'}, \Theta_{\psi_2'}) = z_k) = \begin{cases} c_k & \text{if} \quad k = 1 \\ c_k - c_{k-1} & \text{if} \quad k > 1 \end{cases} \tag{9}$$

This approach allows for the inclusion of atoms with relatively low probability, preventing the Q value from being overestimated. For the transition sample $\boldsymbol{t} = (\boldsymbol{x}, A, r, \boldsymbol{x}')$ and the $i^{\text{th}}$ value atom, the Bellman Operation is as follows:

$$P(\Phi \hat{\mathcal{T}} \tilde{Z}(\boldsymbol{x}, A|\Theta_{\psi_1'}, \Theta_{\psi_2'}) = z_i) = \sum_{j=1}^N \left[ 1 - \frac{|[\hat{\mathcal{T}} z_j]_{V_{MIN}}^{V_{MAX}} - z_i|}{\triangle z} \right]_0^1 P(\tilde{Z}(\boldsymbol{x}', \hat{A}'|\Theta_{\psi_1'}, \Theta_{\psi_2'}) = z_j) \tag{10}$$

where $A$ notes the actual action taken rather than a probability distribution, and $\Phi \hat{\mathcal{T}} \tilde{Z}(\boldsymbol{x}, A|\Theta_{\psi_1'}, \Theta_{\psi_2'})$ is the corrected discrete value distribution used in training $\Theta_{\psi_1}$ and $\Theta_{\psi_2}$.

### 3.4 CRITIC LEARNING

After clarifying the clipped double Q-learning mechanism for discrete value distributions, we will further elaborate on how the dubl critic network learns based on the aforementioned corrected value distributions, so as to achieve accurate estimation of action values and lay the foundation for subsequent strategy optimization.

Because the action that the agent is about to perform is sampled from the distribution. To accommodate this, we developed an updating mechanism for the critic network informed by the previously introduced $\Phi\hat{\mathcal{T}}\tilde{Z}(\boldsymbol{x}, A|\Theta_{\psi'_1}, \Theta_{\psi'_2})$.

$$
\begin{aligned}
P(\Phi\hat{\mathcal{T}}\tilde{Z}(\boldsymbol{x}, \hat{A}|\Theta_{\psi'_1}, \Theta_{\psi'_2}) = z_j) &= \sum_{\forall A \in \mathcal{A}} P(A|\hat{A})P(\Phi\hat{\mathcal{T}}\tilde{Z}(\boldsymbol{x}, A|\Theta_{\psi'_1}, \Theta_{\psi'_2}) = z_j) \\
&\approx \sum_{\boldsymbol{t} \sim \mathcal{D}} P(A|\hat{A})P(\Phi\hat{\mathcal{T}}\tilde{Z}(\boldsymbol{x}, A|\Theta_{\psi'_1}, \Theta_{\psi'_2}) = z_j)
\end{aligned}
\tag{11}
$$

Where $\forall A \in \mathcal{A}$ signifies the requirement to consider every available action within the action space for a perfect estimation, whereas $\boldsymbol{t} \sim \mathcal{D}$ represents the extraction of actions from the replay buffer for an approximation. The first line of the above formula embodies the exhaustive consideration of the action space, where each action's value distribution is aggregated based on its respective occurrence likelihood $P(A|\hat{A})$, constituting the expected value of the distribution across $\hat{A}$. Nevertheless, due to the extensive action space, such exhaustive consideration is impractical. Therefore, we invoke a second-tier approximation by sampling the observed data from the replay buffer, circumventing the full traversal of the action space, notwithstanding the potential distribution bias of the data within the replay buffer. Accordingly, the critic network's update rule for a data batch with size $B$ and $i = 1, 2$ is defined as:

$$
\begin{aligned}
Z_1 &\overset{D}{=} \Phi\hat{\mathcal{T}}\tilde{Z}(\boldsymbol{x}, A|\Theta_{\psi'_1}, \Theta_{\psi'_2}) \\
Z_2 &\overset{D}{=} Z(\boldsymbol{x}, \hat{A}|\Theta_{\psi_i}) \\
\boldsymbol{\psi_i} &\leftarrow \boldsymbol{\psi_i} - \frac{\alpha}{B}\sum_{\boldsymbol{t} \sim \mathcal{D}} P(A|\hat{A})\nabla_{\boldsymbol{\psi_i}}D_{KL}(Z_1||Z_2)
\end{aligned}
\tag{12}
$$

where $D_{KL}$ represents KL divergence, furthermore:

$$
\nabla_{\boldsymbol{\psi_i}}D_{KL}(Z_1||Z_2) = -\sum_{j=1}^{N} P(Z_1 = z_j)\nabla_{\boldsymbol{\psi_i}}\log P(Z_2 = z_j)
\tag{13}
$$

In the above equation, we eliminate terms that are independent of $\boldsymbol{\psi_i}$, thus obtaining a form consistent with cross-entropy loss. $Z_1$ denotes the new estimation of the value distribution procured from the twin critic networks, and $Z_2$ is the critic network's resultant output. In this way, every critic's output is refined to align with the corrected value $Z_1$, reducing the overestimation bias.

### 3.5 POLICY LEARNING

Having introduced how the critic network learns based on the corrected value distributions, this chapter focuses on the training mechanism of the Actor, which is responsible for generating the actions that the critic evaluates. To train the actor robustly, the actor is trained with a loss function similar for training the critic networks as introduced in Section "Critic Learning". The core is to guide policy optimization through value distribution, enabling the Actor to maximize the selection probability of high-value actions while ensuring training stability, enable agents to learn more extensively and make richer and bolder decisions when facing complex HRHR scenarios

Like other RL models with actor-critic architecture, the actor is updated to maximize the Q-value predicted by a critic network. Differently, in our model, the output of the critic networks is probabilistic, so the cumulative distribution can be used as an objective. More specifically, for the $k^{\text{th}}$ value atom,

$$
\begin{aligned}
P(Z(\boldsymbol{x}, \pi_\phi(\boldsymbol{x})|\Theta_{\psi_1}) \in \{z_1, z_2, \cdots z_k\}) &\to 0, \\
P(Z(\boldsymbol{x}, \pi_\phi(\boldsymbol{x})|\Theta_{\psi_1}) \in \{z_{k+1}, z_{k+2}, \cdots z_N\}) &\to 1.
\end{aligned}
\tag{14}
$$

The notation "$\to$" here denotes a trend or movement toward a value. The goal is for the policy to minimize the probability of $Z$ occurring at lower-value atoms while maximizing it at higher-value atoms. With the binary cross-entropy loss applied, the Policy Learning rules are established thus:

$$
\phi \leftarrow \phi + \frac{\alpha}{B}\sum_{\boldsymbol{t} \sim \mathcal{D}}\sum_{j=1}^{N}\nabla_\phi[0\log\rho_j + 1\log(1 - \rho_j)] = \phi + \frac{\alpha}{B}\sum_{\boldsymbol{t} \sim \mathcal{D}}\sum_{j=1}^{N}\nabla_\phi\log(1 - \rho_j)
\tag{15}
$$

where

$$
\rho_j \overset{D}{=} P(Z(\boldsymbol{x}, \pi_\phi(\boldsymbol{x})|\Theta_{\psi_1}) \in \{z_1, z_2, \cdots, z_j\})
\tag{16}
$$

## 3.6 EXPLORATION

Effective exploration is critical in HRHR tasks, as high-reward opportunities may be sparse and require precise maneuvers to discover. A naive or overly broad exploration strategy may never find these solutions. Therefore, we design a heuristic, entropy-based exploration strategy that explicitly links the agent's exploratory behavior to its confidence, as measured by the value distribution, to encourage deeper exploration of promising high-risk regions.

**Definition 3.** *Given an action distribution $\hat{A} = \pi(\boldsymbol{x})$, the action entropy is defined as:*

$$\mathcal{H}(\hat{A}) \overset{D}{=} -\sum_{i=1}^{n}\sum_{j=1}^{m} p_{ij}(\boldsymbol{x}) \log p_{ij}(\boldsymbol{x}) \tag{17}$$

*Additionally, $\mathcal{H}(\hat{A})$ has a calculable upper bound:*

$$\overline{\mathcal{H}} \overset{D}{=} n \log m \geq \mathcal{H}(\hat{A}) \quad \forall \pi : \mathcal{X} \to \mathbb{R}^{n \times m} \tag{18}$$

Our objective is to correlate the action entropy with confidence levels. Specifically, increase the action entropy $\mathcal{H}(\hat{A})$ when there is a higher probability occurrence at lower discretization atoms within the discrete value distribution. To achieve this, we introduce an entropy exploration term. The proposed update rule for the actor is as follows:

$$\phi \leftarrow \phi + \frac{\alpha\beta}{B} \sum_{\boldsymbol{t}\sim\mathcal{D}} s\nabla_\phi \frac{\mathcal{H}(\pi_\phi(\boldsymbol{x}))}{\overline{\mathcal{H}}}$$

$$s = \begin{cases} 1 & if \quad \max_{1\leq j\leq N} \frac{N-j}{N-1}h\rho_j \geq \frac{\mathcal{H}(\pi_\phi(\boldsymbol{x}))}{\overline{\mathcal{H}}} \\ 0 & \text{otherwise} \end{cases} \tag{19}$$

where $\rho_j$ is same to in Equ equation 16 , $\beta > 0$ is the coefficient for the entropy term, $0 < h \leq 1$ regulates the scale of action entropy. An action entropy threshold, $\frac{N-j}{N-1}h\rho_j$, is assigned to each discrete atom of the value distribution such that the entropy exploration term will only activate when the action entropy $\mathcal{H}(\pi_\phi(\boldsymbol{x}))$ falls below this threshold. This threshold decreases as $j$ increases, which means that atoms of higher values have lower thresholds.

We also use the cumulative distribution $\rho_j$ to represent the confidence level of the agent with respect to the current state $\boldsymbol{x}$. It should be noted that for the $j^{\text{th}}$ value atom of a high-confidence agent, $\rho_j$ should be a small scalar since it represents the probability between the $1^{\text{th}}$ atom and the $j^{\text{th}}$ atom, which is the lower value range. We use $\rho_j$ to correct the action entropy $\mathcal{H}(\pi_\phi(\boldsymbol{x}))$, so the low-confidence agent will increase it to seek various solutions with respect to state $\boldsymbol{x}$, however, the high-confidence one will not. Integrating this with the prior section's material, the comprehensive update rule for the actor is:

$$\phi \leftarrow \phi + \frac{\alpha}{B} \sum_{\boldsymbol{t}\sim\mathcal{D}}\sum_{j=1}^{N} \nabla_\phi \log(1-\rho_j) + \frac{\alpha\beta}{B} \sum_{\boldsymbol{t}\sim\mathcal{D}} s\nabla_\phi \frac{\mathcal{H}(\pi_\phi(\boldsymbol{x}))}{\overline{\mathcal{H}}} \tag{20}$$

## 4 EXPERIMENTS

We trained our model on continuous control across multiple tasks using multiple random seeds, including BipedalWalkerHardcore-v3 , FetchPush-v4 and MuJoCo tasks, and evaluated the performance. We also used C51, SAC, SAC-Discrete, TD3, and TQC as baselines. For further implementation details of the experiments, such as, ablation experiment, and detailed description of the environments, please refer to Section 6.4 and 6.5 of the Appendix.

### 4.1 BIPEDALWALKERHARDCORE-V3

The BipedalWalkerHardcore-v (Towers et al., 2023) task is to control the joints of a planar bipedal robot to walk through complex terrains involving randomly generated obstacles like staircase, obstacles, and traps. An agent must attempt to overcome various barriers to achieve the highest possible

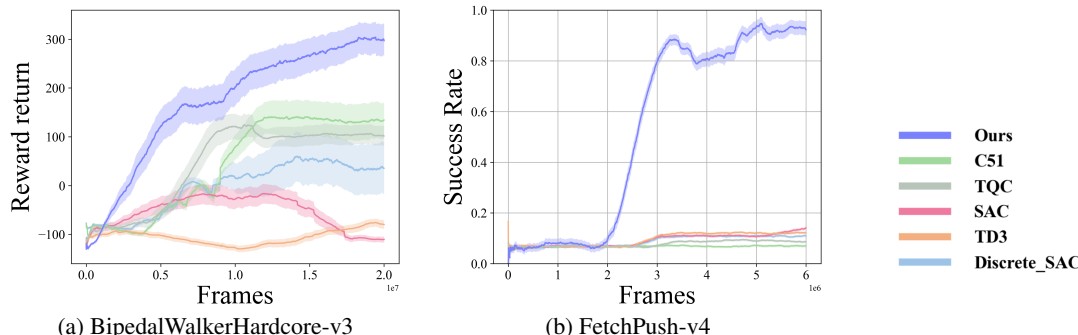

Figure 3: Training Curves for BipedalWalkerHardcore-v3 Experiments and FetchPush-v4

Table 1: Algorithm Performance Comparison Across Environments. (The metric for BipedalWalker is Average Score, while the metric for FetchPush is Success Rate)

| Environment | Ours | C51 | TQC | SAC | TD3 | Discrete-SAC |
|---|---|---|---|---|---|---|
| BipedalWalkerHardcore-v3 | $327.1 \pm 16.1$ | $187.5 \pm 32.4$ | $150.2 \pm 10.8$ | $5.1 \pm 0.7$ | $-20.1 \pm 2.2$ | $82.8 \pm 10.1$ |
| FetchPush-v4 | $0.97 \pm 0.09$ | $0.03 \pm 0.0$ | $0.11 \pm 0.01$ | $0.18 \pm 0.04$ | $0.16 \pm 0.04$ | $0.15 \pm 0.05$ |

score. The challenge lies in its high risk, characterized by randomly varying terrain, partial observability, and high penalties for falling. (Wei & Ying, 2021; Fujimoto et al., 2018)

We trained D2C-HRHR and baselines on the BipedalWalkerHardcore-v3 task for 20 million time steps. Figure 3 shows the reward returns during training. In tests with 10 different random seeds, our model achieved a mean score of 327.1 in 10,000 trials, as shown in Table 1. The experimental results show that TQC, C51 perform better than TD3 and SAC, while our algorithm is the best.

In specific scenarios of the BipedalWalker task, successful decisions yield high rewards, while failed actions result in high penalties, i.e., HRHR scenarios. In a fully observable and deterministic task, TD3 and SAC could distinguish differences in actions and states to fit them well with a scalar expectation. However, in this task, the scalar expectation can be misleading and captures neither the high return nor high risk, but the average. We have verified this in the Appendix 6.3.1. TQC algorithm also have its drawbacks. Although his critic network can output vectors of Q-value distribution, it still uses Gaussian distribution process, which still limits its performance in action exploration.

While C51 utilizes discrete exploration and can capture bimodal distributions (performing stably on stairs), it fails in high-risk scenarios involving stumps or traps. As shown in Fig 4, only our algorithm maintains a bimodal distribution across all obstacle types.

The process of going up and down stairs involves low risk; Even if the agent loses balance, it will only incur a small score deduction. In such cases, C51 which models the reward distribution using a single distribution critic, is less affected by overestimation bias. It can capture the bimodal distribution and achieve good performance. However, in HRHR scenarios such as traps and obstacles, the single critic of the C51 algorithm lacks cross-validation from another critic, making it overestimating the Q-value of certain erroneous actions. As shown **6.3.2** in Appendix, we will demonstrate the key differences between C51 and ours in predicting Q values.

Meanwhile, the Actor in the C51 algorithm only outputs the action atom with the highest probability in the discrete space, rather than sampling outputs based on probability distributions like we do. Our model will enable agents to use more diverse strategies, enabling them to perform better in extreme HRHR environments.

To understand the necessity of each module, we conducted an ablation study on our model for the BipedalWalkerHardcore-v3 task, as shown in Appendix 6.4.1. It was conducted on the Dual Critic Network, Actor, and exploration mechanism to validate the necessity of each module and its impact on performance.

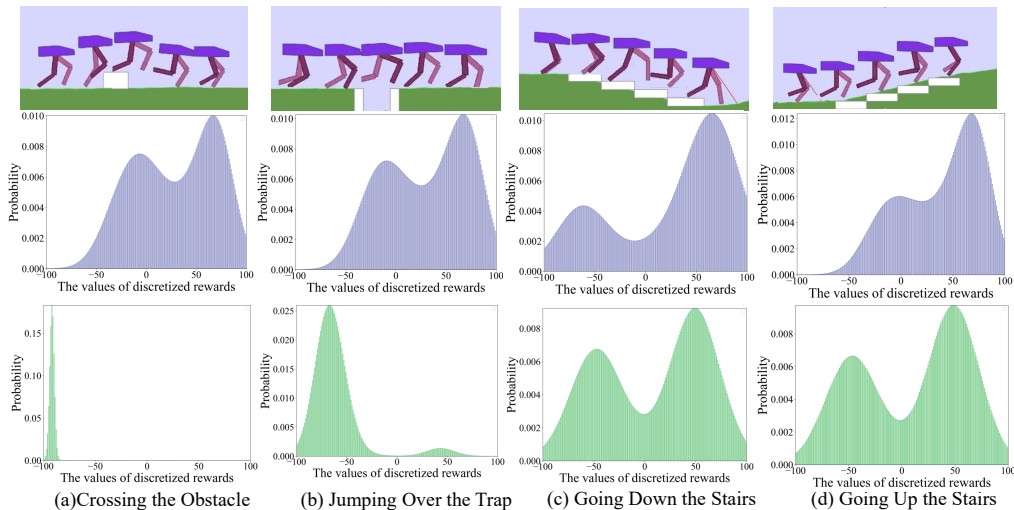

Figure 4: The Distribution Plots of Reward Returns in BipedalWalkerHardcore-v3. ( Blue representing our algorithm and green representing the C51 algorithm. )

## 4.2 FETCHPUSH-V3

To further validate the robustness of the D2C-HRHR model, we conducted tests on the FetchPush-v3 task (Plappert et al., 2018). This task are based on a Fetch robotic arm with 7 degrees of freedom and two parallel grippers, The robotic arm needs to learn to move a block to the target position on the desktop. The difficulty of this task lies in its complex action space.

We conduct 100 tests on each model for each training step to check its success rate and obtain the training curve, as shown in Figure 3(b). Finally, after 5 million steps of training, our algorithm was able to adapt to the contact-rich operating environment and achieved a success rate of 0.97, as shown in the table 1. This demonstrates that our task can adapt to complex action spaces.

## 4.3 MUJOCO ENVIRONMENT

Although the model is intended for tasks with HRHR actions, we also evaluated it in typical continuous control tasks. Experiments were conducted within Mujoco Environment (Towers et al., 2023) on a series of tasks, Ant-v5, HalfCheetah-v5, Hopper-v5, Humanoid-v5, and Walker2D-v5 (Tassa et al., 2012). These tasks involve controlling different types of robots to move forward. Our model and baselines were applied to these tasks and trained over 20 million time steps for each task. Our algorithm ranks second in the total score, as shown in Figure 11, Table 4 and Table 5 in Appendix.

It is worth noting that our model demonstrates particularly outstanding performance on the Humanoid task. This task aims to enable robots to mimic human walking by moving forward as quickly as possible. We observe that robots guided by our algorithm exhibit greater joint flexibility and wider range of motion during running in this task—in other words, they move more like a human. This demonstrates that the Actor in D2C-HRHR enables agents to learn more broadly. Detailed training curves and analysis are provided in Appendix 6.4.2 and Figure12.

## 5 DISCUSSION AND CONCLUSION

In this paper, we propose a distributed reinforcement learning model named D2C-HRHR. It adopts a discrete action space, and employs a unique clipped dobule Q learning approach, policy learning based on discrete action probability distribution sampling, and a cross-entropy nested exploration mechanism. This model demonstrates outstanding performance in HRHR scenarios, achieving capabilities unmatched by other baselines. It solves BipedalWalkerHardcore-v3 with state-of-the-art performance and exhibits excellent performance in various continuous control tasks.

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

## ETHICS STATEMENT

This work focuses on methodological and theoretical advances in discrete reinforcement learning. It does not involve human or animal subjects, nor does it rely on sensitive or proprietary data. We do not anticipate any immediate ethical concerns. Potential applications of reinforcement learning should always be carefully evaluated to prevent harmful or unsafe use.

## REPRODUCIBILITY STATEMENT

We provide detailed descriptions of our algorithm, theoretical definitions and proofs, and experimental setups to ensure reproducibility. All assumptions and complete proofs of theoretical results are included in the Appendix.

For experiments, we describe setup and implementation details in the Appendix and Supplementary Material. Results and analysis of experiments are provided in the main text and appendix. Experiments are conducted on a desktop workstation with the Intel® Core TMi9-12900 Processor, 64GB RAM, and the NVIDIA® GeForce RTX TM4090. The code for the algorithm can be found in the additional materials. Below are the hyperparameters used by our algorithm in various environments.

We plan to release our source code publicly upon acceptance of the paper. We believe these resources will help other researchers to reproduce our findings.

Table 2: Training Results Comparison

| Environment | Hyperparameters | | | | |
|---|---|---|---|---|---|
| | **Learning Rate** | $V_{Min}$ | $V_{Max}$ | $\gamma$ | Batch Size |
| BipedalWalkerHardcore-3 | $2.5 \times 10^{-4}$ | -100 | 100 | 0.99 | 512 |
| FetchPush-v4 | $3 \times 10^{-5}$ | -50 | 50 | $1 - 1/50$ | 512 |
| Mujuco | $1 \times 10^{-4}$ | -200 | 200 | 0.99 | 1024 |

# 6 APPENDIX

## 6.1 MATH SYMBOLS

Table 3: Mathematical Symbols

| Symbol | Description | Typical value |
|---|---|---|
| $\gamma$ | Discount factor. | 0.98 or 0.99 |
| $V_{MAX}$ | Upper bound of the discrete value. | $\frac{1}{1-\gamma}$ |
| $V_{MIN}$ | Lower bound of discrete value. | $-\frac{1}{1-\gamma}$ |
| $Z$ | Random variable for discrete value. | |
| $z_i$ | The $i^{\text{th}}$ discrete value atom of $Z$. | $[V_{MIN}, V_{MAX}]$ |
| $\boldsymbol{x}$ | Sample of current state observation. | |
| $\boldsymbol{a}$ | An action sample vector. | |
| $\hat{\boldsymbol{a}}$ | Distribution of an action vector. | |
| $\hat{A}$ | An action distribution matrix of a multidimensional discrete action space, in which the sum of each row is 1. | |
| $A$ | An action sample matrix. Each row in $A$ adopts one-hot coding sampled from the corresponding row in $\hat{A}$. | |
| $\hat{A}'$ | Action distribution for the next state. | |
| $r$ | Reward from an environment. | $\leq 1$ |
| $\boldsymbol{x}'$ | Sample of next state observation. | |
| $\hat{\boldsymbol{x}}'$ | Distribution of the next state observation. | |
| $\hat{\cdot}$ | Distribution of a random variable. | |
| $\cdot'$ | A variable in the next time step. | |
| $\overset{D}{=}$ | Denotes definition. | |
| $\mathcal{X}$ | Space of state observations. | |
| $\mathcal{A}$ | Space of actions. | |
| $\left\lceil \frac{1}{1-\gamma} \right\rceil$ | Ceiling of $\frac{1}{1-\gamma}$. | |
| $N$ | Number of discrete atoms for $Z$. | 51 |
| $\hat{\mathcal{T}}Z$ | $r + \gamma Z'$. | |
| $\Phi\hat{\mathcal{T}}Z$ | Projecting $\hat{\mathcal{T}}Z$ back to origin discrete value atoms. | |
| $\tilde{Z}$ | The estimated Z from the twin critic networks. | |
| $\Theta$ | Discrete distribution critic network. | |
| $(\cdot)_i$ | $i^{\text{th}}$ Element of a Vector. | |
| $\pi$ | Policy for action selection. | |
| $Q$ | Expected scalar critic network. | |
| $\boldsymbol{\psi_1}, \boldsymbol{\psi_2}$ | Parameters of first and second critic networks. | |
| $\boldsymbol{\phi}$ | Parameters of actor network | |
| $\boldsymbol{\psi_1'}, \boldsymbol{\psi_2'}, \phi'$ | Parameters for delayed updated networks. | |
| $\leftarrow$ | Denotes parameter update. | |
| $\nabla\boldsymbol{\omega} J$ | Gradient of $J$ with respect to $\boldsymbol{\omega}$. | |
| $\alpha$ | Learning rate. | $\leq 10^{-3}$ |
| $B$ | Batch size. | 256 or 512 |
| $\boldsymbol{t} \sim \mathcal{D}$ | Sample from Replay Buffer. | |
| $n$ | Number of dimensions in action. | $\leq 20$ |
| $m$ | Number of atoms per action dimension. | 51 |
| $\mathcal{H}(\hat{A})$ | Entropy of action $\hat{A}$. | $\leq n \log m$ |
| $\overline{\mathcal{H}}$ | Maximum entropy of action. | $n \log m$ |
| $h$ | Scaling factor for action entropy. | 0.5 |
| $\beta$ | Coefficient for exploration. | 0.5 |
| $\sup$ | Represents the upper bound. | |
| $\Omega_1$ | The high-risk-high-return region. | |
| $\Omega R_2$ | The low-risk-stable-return region. | |
| $\Omega R_i$ | A generalized notation for any subregion of the action space, used for mathematical uniformity. | |

## 6.2 Further background

### 6.2.1 Fundamentals of Distributed Reinforcement Learning

This chapter will introduce some basic concepts regarding the model we proposed. Reading this chapter will help you gain some basic knowledge about discrete reinforcement learning. Our model is extended on the basis of the content of this chapter.

For a stochastic transition process $(\boldsymbol{x}, \boldsymbol{a}) \to (\hat{\boldsymbol{x}}', \hat{\boldsymbol{a}}')$ in a environment, $\boldsymbol{x}$ represents the observed current state of the environment, and $\boldsymbol{a}$ specifies the action taken in response to $\boldsymbol{x}$. The resulting state distribution is denoted $\hat{\boldsymbol{x}}'$. A stochastic policy output an action distribution $\hat{\boldsymbol{a}}'$, and the actual action $\boldsymbol{a}'$ taken in the task will be sampled from $\hat{\boldsymbol{a}}'$.

The value $Z$ is a discrete distribution and is associated with the process $(\boldsymbol{x}, \boldsymbol{a}) \to (\hat{\boldsymbol{x}}', \hat{\boldsymbol{a}}')$. It can be formularized using a recursive equation:

$$Z(\boldsymbol{x}, \boldsymbol{a}) \overset{D}{=} R(\boldsymbol{x}, \boldsymbol{a}) + \gamma Z(\hat{\boldsymbol{x}}', \hat{\boldsymbol{a}}') \tag{21}$$

wherein $R(\boldsymbol{x}, \boldsymbol{a})$ represents the stochastic reward function of the environment and $\gamma$ denotes the discount rate.

Subsequently, the value $Z$ is conceptualized as a random variable with a discrete value distribution. The number of discrete atoms $N \in \mathbb{N}$ denotes the granularity of discretization required for the value domain, and the bounds $V_{MIN}, V_{MAX} \in \mathbb{R}$ specify the lower and upper limits of the values, respectively. The set of discrete atoms is constructed as $\{z_i = V_{MIN} + (i-1)\triangle z | i = 1, 2, \cdots N\}$, with the interval $\triangle z$ calculated by $\frac{V_{MAX} - V_{MIN}}{N-1}$. The probability of each discrete atom's occurrence is determined using a neural network $\Theta : \mathcal{X} \times \mathcal{A} \to \mathbb{R}^N$.

$$Z(\boldsymbol{x}, \boldsymbol{a}|\Theta) = z_i \quad w.p. \quad p_i(\boldsymbol{x}, \boldsymbol{a}) = \frac{e^{(\Theta(\boldsymbol{x}, \boldsymbol{a}))_i}}{\sum_j^N e^{(\Theta(\boldsymbol{x}, \boldsymbol{a}))_j}} \tag{22}$$

For a tuple of a stochastic transition $\boldsymbol{t} = (\boldsymbol{x}, \boldsymbol{a}, r, \boldsymbol{x}')$, a Bellman update for a discrete distribution is applied to each discrete atom $z_j$, designated as $\hat{\mathcal{T}} z_j := r + \gamma z_j$. The probability associated with $\hat{\mathcal{T}} z_j$, denoted $p_j(\boldsymbol{x}', \pi(\boldsymbol{x}'))$, is then redistributed amongst adjacent discrete atoms. The $i^{\text{th}}$ element of the resultant projected discrete probability distribution $\Phi\hat{\mathcal{T}} Z(\boldsymbol{x}, \boldsymbol{a}|\Theta)$ is:

$$P(\Phi\hat{\mathcal{T}} Z(\boldsymbol{x}, \boldsymbol{a}|\Theta) = z_i) \quad = \sum_{j=1}^{N} \left[ 1 - \frac{|[\hat{\mathcal{T}} z_j]_{V_{MIN}}^{V_{MAX}} - z_i|}{\triangle z} \right]_0^1 p_j(\boldsymbol{x}', \pi(\boldsymbol{x}'))$$

The notation $[\cdot]_a^b$ signifies that the value is constrained within the interval $[a, b]$.

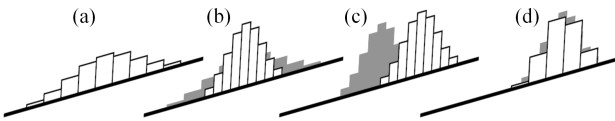

Figure 5: Operations to update $Z$. (a) The current value distribution of $Z$. (b) Discount factor $\gamma$ changes the shape in the dimension of atoms. (c) The current reward $R$ shifts the distribution in the dimension of atoms. (d) The resulting distribution $R + \gamma Z$ is mapped back to the original atoms by $\Phi$.

### 6.2.2 ALGORITHM OF DOUBLE CRITIC NETWORK

---

**Algorithm 1** Dual-Critic Network Based on Discrete Value Distribution

---

1: **Input:** Twin critic networks $\Theta_{\psi_1'}$, $\Theta_{\psi_2'}$, next state $\boldsymbol{x'}$
2: **Output:** Refined value distribution $\tilde{Z}(\boldsymbol{x'}, \hat{A}')$

3: **procedure** DUALCRITICEVALUATION
4:     $\Phi_1, \Phi_2 \leftarrow$ Estimate discrete value distributions for $\boldsymbol{x'}$ using both critics
5:     $C_1, C_2 \leftarrow$ Compute cumulative distributions from $\Phi_1$ and $\Phi_2$
6:     **for** each value category $k = 1$ to $N$ **do**
7:         $c_k \leftarrow \max(C_1[k], C_2[k])$          ▷ Select conservative cumulative probability
8:     **end for**
9:     **for** each value category $k = 1$ to $N$ **do**
10:         **if** $k = 1$ **then**
11:             $\tilde{P}_k \leftarrow c_1$
12:         **else**
13:             $\tilde{P}_k \leftarrow c_k - c_{k-1}$          ▷ Convert back to probability distribution
14:         **end if**
15:     **end for**
16:     **return** $\tilde{Z}$ with probabilities $\tilde{P}_1, \tilde{P}_2, \ldots, \tilde{P}_N$
17: **end procedure**

---

### 6.3 A DEEPER ANALYSIS AND ILLUSTRATION OF MECHANISMS

#### 6.3.1 THE REASON WHY A POLICY WITH GAUSSIAN ACTIONS PERFORMS WORSE IN HRHR SCENARIOS

Section 3.1 introduces Theorem 1. Here we recall to it again and prove it:

For Gaussian policy $\pi_\theta(\cdot|s) = \mathcal{N}(\mu_\theta(s), \sigma_\theta^2(s)I)$ in an HRHR scenario at state $s$ with high-reward grains $\mathbb{G}_1$ of diameter $\delta$, if $\sigma_\theta(s) > \delta$, then the gradient update satisfies:

$$\langle \nabla_\theta J(\theta), \Delta\theta_{R_1} \rangle < \langle \nabla_\theta J(\theta), \Delta\theta_{R_2} \rangle \tag{23}$$

where $\Delta\theta_{R_i}$ is the update direction toward region $R_i$, and $J(\theta)$ is the expected return. This implies gradient updates prefer $R_2$ over $R_1$.

*Proof.* The policy gradient is:

$$\nabla_\theta J(\theta) = \mathbb{E}_{a \sim \pi_\theta}[\nabla_\theta \log \pi_\theta(a|s) Q(s,a)] \tag{24}$$

For Gaussian policies, the score function is:

$$\nabla_{\mu_\theta} \log \pi_\theta(a|s) = \sigma_\theta^{-2}(s)(a - \mu_\theta(s)) \tag{25}$$

The key inner product is:

$$\langle \nabla_\theta J(\theta), \Delta\theta_{R_i} \rangle = \mathbb{E}_{a \sim \pi_\theta}[\langle \Delta\theta_{R_i}, \nabla_\theta \log \pi_\theta(a|s) Q(s,a) \rangle] \tag{26}$$

$$= \sigma_\theta^{-2}(s) \mathbb{E}_{a \sim \pi_\theta}[\langle \Delta\theta_{R_i}, (a - \mu_\theta(s)) \rangle Q(s,a)] \tag{27}$$

Define the advantage relative to $R_2$:

$$A_{R_2}(s,a) = Q(s,a) - \mathbb{E}_{a' \sim \mathcal{U}(R_2)}[Q(s,a')] \tag{28}$$

The difference in update directions is:

$$\langle \nabla_\theta J(\theta), \Delta\theta_{R_1} - \Delta\theta_{R_2} \rangle \tag{29}$$

$$= \sigma_\theta^{-2}(s) \mathbb{E}_{a \sim \pi_\theta}[\langle \Delta\theta_{R_1} - \Delta\theta_{R_2}, (a - \mu_\theta(s)) \rangle A_{R_2}(s,a)] \tag{30}$$

Under $\sigma_\theta(s) > \delta$, the covariance between action displacement and advantage is:

$$\text{Cov}_{a \sim \pi_\theta}(a - \mu_\theta(s), A_{R_2}(s,a)) < 0 \tag{31}$$

because high-reward grains contribute negligibly due to their small size ($\delta$) relative to policy variance ($\sigma_\theta(s)$). Thus:

$$\langle \nabla_\theta J(\theta), \Delta\theta_{R_1} \rangle < \langle \nabla_\theta J(\theta), \Delta\theta_{R_2} \rangle \tag{32}$$

$\square$

When $\sigma_\theta(s) > \delta$, the policy's exploration radius exceeds high-reward grain sizes. This makes $R_1$'s low average return dominate over its high maximum return, causing gradient updates to prefer $R_2$.

This explains why Gaussian policies with fixed large variance struggle in HRHR scenarios. Adaptive variance schedules or heavy-tailed distributions are often necessary to capture high-reward regions.

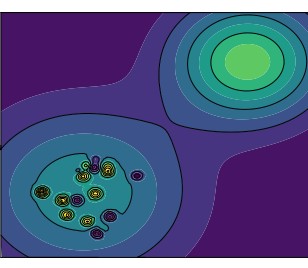 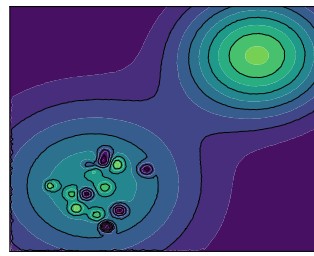

(a) HRHR Scenarios                    (b) Output Scalar Q-value Expectation

Figure 6: Illustration of returns sampled by Gaussian distribution actions. After sampling, the positions where highest true returns locate no longer keep the highest, but the position with the suboptimal true return is highest.

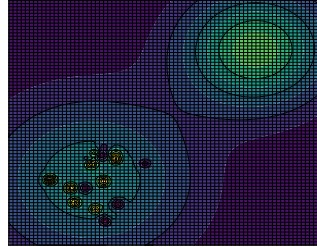 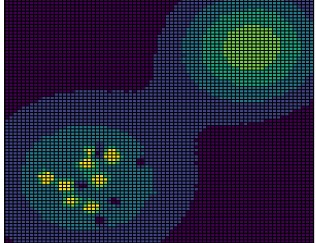

(a) HRHR Scenarios with Discrete Space      (b) Select the Cell with the Highest
                                             Expected Q-value for Output

Figure 7: Illustration of returns sampled by discrete actions.

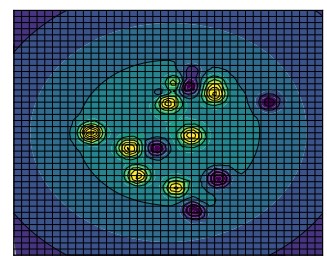 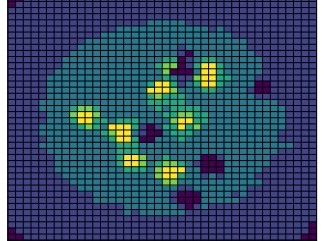

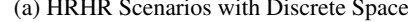

(a) HRHR Scenarios with Discrete Space      (b) Critic 1 Output Q-Value Prediction

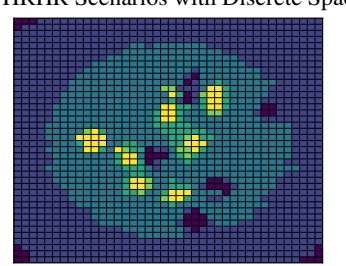 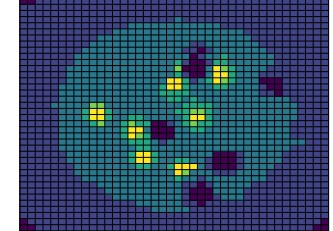

(c) Critic 2 Output Q-Value Prediction      (d) Select Lower Q Value and Output the
                                             Probability Distribution of Action Atoms

Figure 8: Illustration of double discrete critics

### 6.3.2 ILLUSTRATIONS OF Q-VALUE ESTIMATION WITH DIFFERENT ALGORITHMS

For a more intuitive illustration of why a Gaussian policy is hard to find the optimal actions in HRHR scenarios, we plot contour maps to illustrate landscapes of true returns and possible estimated returns after sampling the true returns on Gaussian distribution actions.

As shown in Figure 6, (a) is the true landscape and (b) is the landscape after a Gaussian blur to approximate the sampling of the returns by actions with a Gaussian distribution. Although the best return in (a) is in the bottom left, the best return in (b) is in the top right, which is not optimal. It is the case for algorithms with a policy with Gaussian action, such as SAC and TD3.

Differently, if the action dimensions are discretised (Figure 7 (a)) and the returns are sampled by discrete actions (Figure 7 (b)), although the resolution is much lower, the high return regions in the HRHR scenario are more likely to be captured. It is the case for algorithms with discrete actions.

However, because of the sharp gradient in a box after the discretization, the sampled return is a distribution; hence, adopting a critic with distributional output is beneficial. Like critics with scalar output, we notice critics with distributional outputs also suffer from overestimation; hence, in our model, we mitigate it by using double critics. Figure 8 shows an example of discrete double critics by a zoom-in of the return landscape. (b) and (c) show two samples by discrete actions which illustrate the estimation of two critics, (d) shows choosing the lower returns from (b) and (c) and combining them for less overestimated returns.

### 6.3.3 TRAP CHEESE PROBLEM AND MATHEMATICAL ANALYSIS

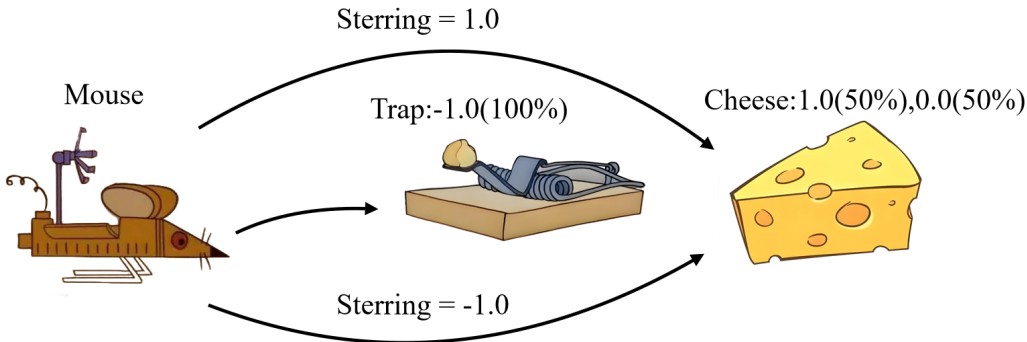

Figure 9: Trap Cheese Problem

We designed a toy task called "Trap or Cheese" to illustrate that continuous models averaging good actions can result in a bad action, but our model does not have this problem. As shown in Figure 9, there is a trap in front of the mouse, behind which lies a piece of cheese. When the mouse chooses to move straight ahead, it falls into the trap and dies, resulting in a reward of -1.0. When the mouse chooses to turn left or right, it can bypass the trap and reach the cheese. However, there is a 50% chance that the cheese has expired and cannot be eaten, resulting in a reward of 0.0. If the cheese is not expired, the reward is 1.0. Obviously, a normal mouse would not choose to walk into the trap.

Both SAC and our model are tested in this task. The results show that SAC tends to unhesitatingly choose the middle route and walk into the trap, with an average score staying at -1.0. In contrast, our discrete model can learn the correct strategy, with an average score staying at 0.5. This simple task is difficult for SAC because, although its critic network can learn that moving forward is a very bad choice, since moving forward can be considered as an average of moving left and right, SAC still chooses to move forward. This problem could widely exist in continuous RL models which tend to average best actions. The BipedalWalkerHardcore task shares a similar property when stepping over obstacles. Hence, we suspect it is the reason why continuous models cannot solve this task as well as our model. For further discussion and mathematical analysis, please refer to the following proof.

We describe the $Q$ function of the Trap Cheese problem as:

$$Q(x_0, a) = \begin{cases} 0.5 & if \quad a \in [-1-\delta, -1+\delta] \cup [1-\delta, 1+\delta] \\ -1 & otherwise \end{cases} \tag{33}$$

Where $\delta$ is used to denote the width of the range where high rewards can be obtained, $0 < \delta < 1$. For convenience, we represent this region with the symbol $\mathcal{C}(\delta)$. We are interested in the maximum likelihood estimation of the normal distribution $a \sim N(\mu, \sigma^2)$ on $\mathcal{C}(\delta)$.

$$\begin{aligned} \log L &= \int_{\mathcal{C}(\delta)} \log(\frac{1}{\sqrt{2\pi}\sigma} e^{-\frac{(a-\mu)^2}{2\sigma^2}}) da \\ &= -4\delta \log(\sqrt{2\pi}\sigma) - \frac{1}{2\sigma^2} \int_{\mathcal{C}(\delta)} (a-\mu)^2 da \\ &= -4\delta \log(\sqrt{2\pi}\sigma) - \frac{1}{6\sigma^2} [(1+\delta-\mu)^3 - (1-\delta-\mu)^3 + (-1+\delta-\mu)^3 - (-1-\delta-\mu)^3] \\ &= -4\delta \log(\sqrt{2\pi}\sigma) - \frac{1}{6\sigma^2} [6(1-\mu)^2\delta + 2\delta^3 + 6(1+\mu)^2\delta + 2\delta^3] \\ &= -4\delta \log(\sqrt{2\pi}\sigma) - \frac{2\delta}{3\sigma^2} [3(1+\mu^2) + \delta^2] \end{aligned} \tag{34}$$

Letting $\frac{\partial \log L}{\partial \mu} = 0$ and $\frac{\partial \log L}{\partial \sigma} = 0$, we can obtain the maximum likelihood estimates for $\mu$ and $\sigma$.

$$\begin{aligned} \frac{\partial \log L}{\partial \mu} &= -\frac{4\delta\mu}{\sigma^2}, \quad \tilde{\mu} = 0 \\ \frac{\partial \log L}{\partial \sigma} &= -\frac{4\delta}{\sigma} + \frac{4\delta}{3\sigma^3}[3(1+\mu^2) + \delta^2], \quad \tilde{\sigma}^2 = 1 + \frac{\delta^2}{3} \end{aligned} \tag{35}$$

Hessian matrix helps to verify whether $\tilde{\mu} = 0$ and $\tilde{\sigma}^2 = 1 + \frac{\delta^2}{3}$ is the unique critical point.

$$\begin{bmatrix} \frac{\partial^2 \log L}{\partial \mu^2} & \frac{\partial^2 \log L}{\partial \mu \partial \sigma} \\ \frac{\partial^2 \log L}{\partial \mu \partial \sigma} & \frac{\partial^2 \log L}{\partial \sigma^2} \end{bmatrix} = \begin{bmatrix} -\frac{4\delta}{\sigma^2} & \frac{8\delta\mu}{\sigma^3} \\ \frac{8\delta\mu}{\sigma^3} & \frac{4\delta}{\sigma^4}(\sigma^2 - \delta^2 - 3) \end{bmatrix} \tag{36}$$

Substituting $\tilde{\mu} = 0$ and $\tilde{\sigma}^2 = 1 + \frac{\delta^2}{3}$, we obtain:

$$\begin{bmatrix} \frac{\partial^2 \log L}{\partial \mu^2} & \frac{\partial^2 \log L}{\partial \mu \partial \sigma} \\ \frac{\partial^2 \log L}{\partial \mu \partial \sigma} & \frac{\partial^2 \log L}{\partial \sigma^2} \end{bmatrix}_{\tilde{\mu}, \tilde{\sigma}} = \begin{bmatrix} -\frac{4\delta}{1+\frac{\delta^2}{3}} & 0 \\ 0 & -\frac{8\delta}{1+\frac{\delta^2}{3}} \end{bmatrix} \preceq 0 \tag{37}$$

Therefore, $\tilde{\mu} = 0$ and $\tilde{\sigma}^2 = 1 + \frac{\delta^2}{3}$ is the unique maximum point on the domain. Although $N(\tilde{\mu}, \tilde{\sigma}^2)$ is the maximum likelihood estimate for the set $\mathcal{C}(\delta)$ under the assumption of a normal distribution, its maximum probability density point $\tilde{\mu}$ does not yield satisfactory values on the $Q$ function; Obviously, $Q(x_0, \tilde{\mu}) = -1$. Now we will compute the maximum likelihood estimate again, this time on a discrete distribution.

$$P(a = a_i) = p_i, \quad i = 1, 2, \cdots, m, \quad \sum_i^m p_i = 1.0, \quad p_i >= 0 \tag{38}$$

In the case of a discrete distribution, the range of action $a$ is given by $\mathcal{A} = \{a_1, a_2, \cdots, a_m\}$, and $\mathcal{A} \cap \mathcal{C}(\delta) \neq \emptyset$.

$$L = \prod_{\mathcal{A} \cap \mathcal{C}(\delta)} pi \tag{39}$$

According to AM-GM inequality, we have:

$$\prod_{\mathcal{A} \cap \mathcal{C}(\delta)}^{\|\mathcal{A} \cap \mathcal{C}(\delta)\|} pi \leq \frac{\sum_{\mathcal{A} \cap \mathcal{C}} p_i}{\|\mathcal{A} \cap \mathcal{C}(\delta)\|} \leq \frac{1}{\|\mathcal{A} \cap \mathcal{C}(\delta)\|} \tag{40}$$

The two equalities in the above inequality can be attained; therefore, the maximum likelihood estimate in the case of a discrete distribution is:

$$\tilde{p}_i = \begin{cases} \frac{1}{\|\mathcal{A} \cap \mathcal{C}(\delta)\|} & if \quad a_i \in \mathcal{C}(\delta) \\ 0 & otherwise \end{cases} \tag{41}$$

In the maximum likelihood estimate of a discrete distribution, we take the point $a_k$ with the highest probability, and obviously it satisfies $Q(x_0, a_k) = 0.5$. The above result suggests that when dealing with complex obstacles, discrete distributions might have an advantage over normal distributions, at least in the context of maximum likelihood estimation.

## 6.4 FURTHER EXPERIMENT RESULTS

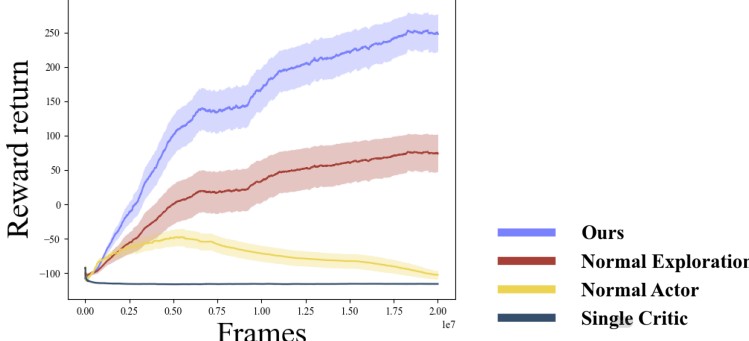

Figure 10: Ablation Experiments of over 10,000 Trials for BipedalWalkerHardcore-v3

### 6.4.1 ABLATION EXPERIMENTS OF D2C-HRHR ON BIPEDALWALKERHARDCORE-V3

As shown in the Figure 10, we conducted ablation experiments on each module of the algorithm in different random seeds. Following D3PG (Barth-Maron et al., 2018), we substituted our discrete actor with a conventional continuous action actor based on our twin critic network. The results are depicted by the curve labeled "Normal Actor". Additionally, we also attempted to replace our exploration mechanism based on action entropy with the exploration mechanism relying on fixed noise from the C51 algorithm, and the results are represented by the curve labeled "Normal Exploration". The results suggest that the different modules proposed in our model are necessary for the model's performance.

### 6.4.2 RESULTS AND ANALYSIS OF THE MUJOCO MISSION

Calculate the test scores of our algorithm and baseline on the training curves of five tasks (Ant-v5, HalfSheetah-v5, Hopper-v5, Humanoid-v5, and Walker2D-v5). The training curves of various algorithms on MuJoCo tasks, as well as the specific scores and standard deviations after 10,000 evaluations, can be found in Figure 11 and Table 4. Assuming that the weights of the five tasks are the same, by taking a weighted average of the scores of each algorithm on different tasks, we find that D2C-HRHR ranks second in the total score, second only to the SAC algorithm, as shown in Table 5.

Our algorithm performs exceptionally well on Humanoid-v5. D2C-HRHR enables humanoid robots to walk with greater amplitude and more adventurous movements. This not only makes the robot walk faster but also more like a real human. As evidence, in Figure 12, we selected three algorithms that performed best in this task for testing, obtaining the joint position distribution maps of the humanoid robot's lower limbs, including the knee and hip joints.

As shown, the robot guided by our algorithm exhibits highly flexible joints during testing, moving in a remarkably fluid manner. In contrast, the SAC and TD3 algorithms produce relatively fixed joint positions, causing the robot to advance in a crawling fashion. Despite this movement style, they still achieve relatively high rewards. As mentioned earlier, the characteristics of D2C-HRHR stem from its unique architecture.

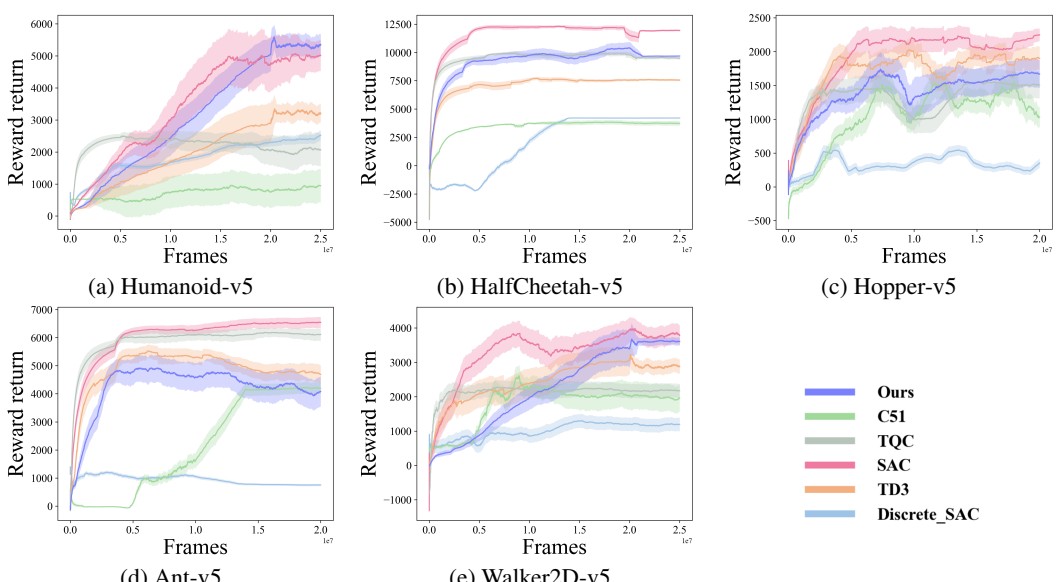

Figure 11: Training Curves for the MuJoCo Environments

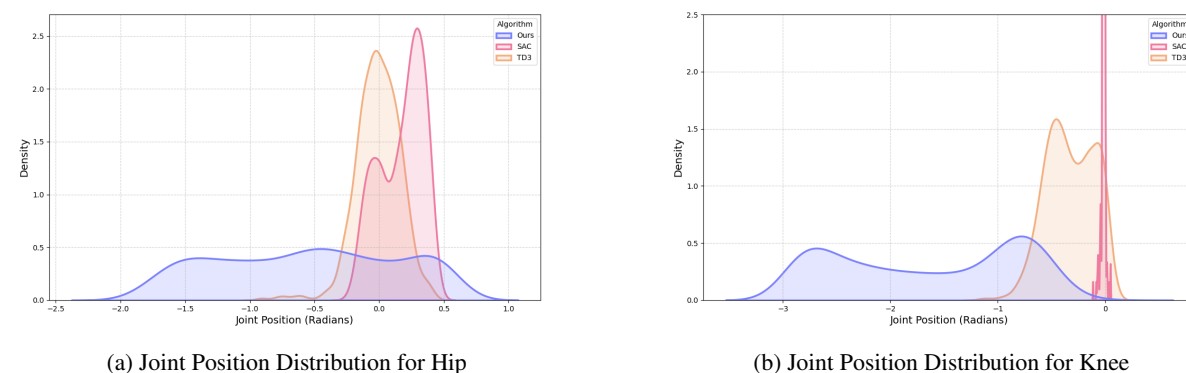

(a) Joint Position Distribution for Hip       (b) Joint Position Distribution for Knee

Figure 12: Joint Position Distributions for the Robot of Task Humanoid-v5 (Blue is Ours, red is SAC, orange is TD3

Table 4: Algorithm Final Evaluation over 10,000 Trials for MuJoCo

| Environment | Average Score ($\pm$ Standard Deviation) | | | | | |
|---|---|---|---|---|---|---|
| | **Ours** | C51 | TQC | SAC | TD3 | Discrete-SAC |
| Humanoid-v5 | **5426.2 $\pm$ 75.3** | 1042.2 $\pm$ 80.1 | 1821.0 $\pm$ 46.4 | 5050.4 $\pm$ 91.0 | 3241.1 $\pm$ 74.6 | 2241.8 $\pm$ 30.7 |
| Ant-v5 | **4468.7 $\pm$ 96.1** | 4471.0 $\pm$ 50.3 | 5821.2 $\pm$ 21.7 | 6001.0 $\pm$ 14.2 | 4521.3 $\pm$ 91.2 | 930.0 $\pm$ 9.4 |
| HalfCheetah-v5 | **10000.8 $\pm$ 37.4** | 2347.2 $\pm$ 17.6 | 9942.6 $\pm$ 32.3 | 11472.6 $\pm$ 28.5 | 3986.0 $\pm$ 22.2 | 2802.3 $\pm$ 10.1 |
| Hopper-v5 | **1745.2 $\pm$ 67.1** | 985.5 $\pm$ 45.2 | 1453.0 $\pm$ 78.8 | 2420.8 $\pm$ 60.6 | 1977.1 $\pm$ 52.2 | 422.8 $\pm$ 18.7 |
| Walker2D-v5 | **3663.6 $\pm$ 78.6** | 2021.8 $\pm$ 62.6 | 2110.0 $\pm$ 36.8 | 3840.5 $\pm$ 55.7 | 2740.2 $\pm$ 77.1 | 1025.4 $\pm$ 37.1 |

Table 5: Overall Performance Comparison of Six Tasks in the Mujoco Series

| Algorithm | Composite Score |
|---|---|
| **Ours** | **4.76** |
| SAC | 5.14 |
| TD3 | 3.55 |
| TQC | 3.36 |
| C51 | 1.87 |
| Discrete-SAC | 1.69 |

### 6.4.3 AN DETAILED DESCRIPTION TO VARIOUS TASKS IN EXPERIMENT

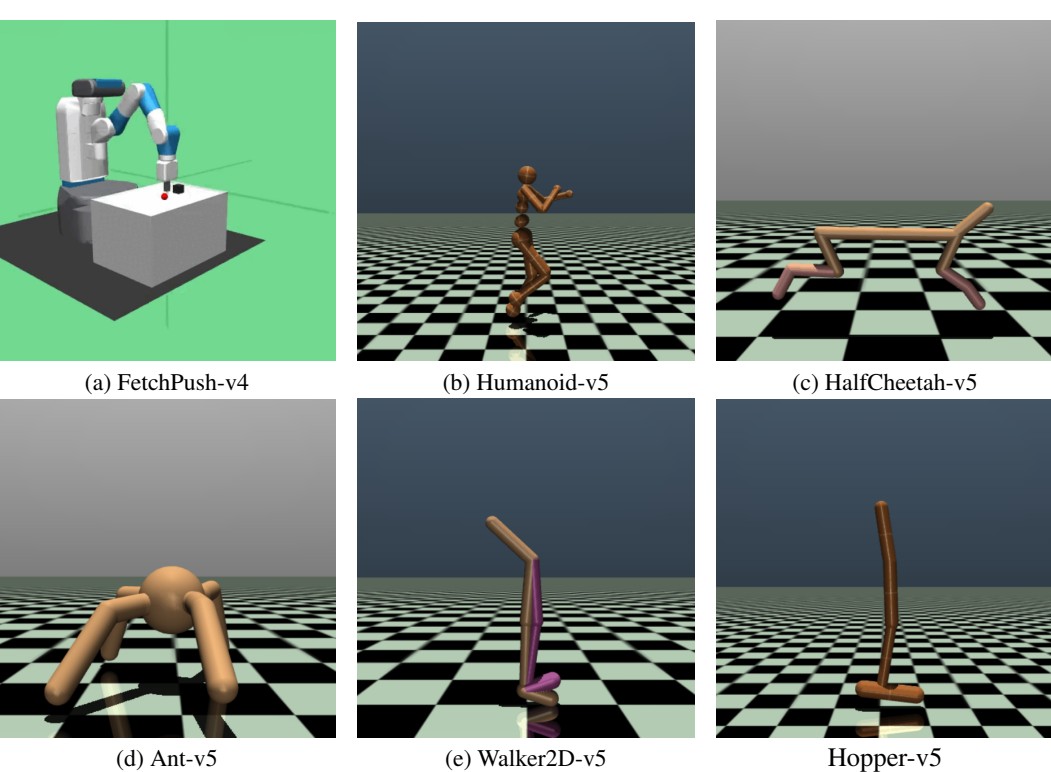

(a) FetchPush-v4    (b) Humanoid-v5    (c) HalfCheetah-v5

(d) Ant-v5    (e) Walker2D-v5    Hopper-v5

Figure 13: Presentation of Various Tasks in the Experiment

As shown in Figure 13.

FetchPush-v4 : The task in the environment is for a manipulator to move a block to a target position on top of a table by pushing with its gripper. The robot is a 7-DoF Fetch Mobile Manipulator with a two-fingered parallel gripper. The robot is controlled by small displacements of the gripper in Cartesian coordinates and the inverse kinematics are computed internally by the MuJoCo framework. The gripper is locked in a closed configuration in order to perform the push task. The task is also continuing which means that the robot has to maintain the block in the target position for an indefinite period of time.

Ant-v5: The ant is a 3D quadruped robot consisting of a torso (free rotational body) with four legs attached to it, where each leg has two body parts. The goal is to coordinate the four legs to move in the forward (right) direction by applying torque to the eight hinges connecting the two body parts of each leg and the torso (nine body parts and eight hinges).

Humanoid-v5: The 3D bipedal robot is designed to simulate a human. It has a torso (abdomen) with a pair of legs and arms, and a pair of tendons connecting the hips to the knees. The legs each consist

of three body parts (thigh, shin, foot), and the arms consist of two body parts (upper arm, forearm). The goal of the environment is to walk forward as fast as possible without falling over.

HalfCheetah-v5: The HalfCheetah is a 2-dimensional robot consisting of 9 body parts and 8 joints connecting them (including two paws). The goal is to apply torque to the joints to make the cheetah run forward (right) as fast as possible, with a positive reward based on the distance moved forward and a negative reward for moving backward. The cheetah's torso and head are fixed, and torque can only be applied to the other 6 joints over the front and back thighs (which connect to the torso), the shins (which connect to the thighs), and the feet (which connect to the shins).

Walker2D-v5: Like other MuJoCo environments, this environment aims to increase the number of independent state and control variables compared to classical control environments. The walker is a two-dimensional bipedal robot consisting of seven main body parts - a single torso at the top (with the two legs splitting after the torso), two thighs in the middle below the torso, two legs below the thighs, and two feet attached to the legs on which the entire body rests. The goal is to walk in the forward (right) direction by applying torque to the six hinges connecting the seven body parts.

Hopper-v5: The environment aims to increase the number of independent state and control variables compared to classical control environments. The hopper is a two-dimensional one-legged figure consisting of four main body parts - the torso at the top, the thigh in the middle, the leg at the bottom, and a single foot on which the entire body rests. The goal is to make hops that move in the forward (right) direction by applying torque to the three hinges that connect the four body parts.

## 6.5 IMPLEMENTATION DETAILS

### 6.5.1 REWARD NORMALIZATION

Reward Normalization is crucial in the training and convergence of models. The original reward function, denoted as $R_1(\boldsymbol{x}, A)$, is advised to be transformed into a normalized form $R_2(\boldsymbol{x}, A)$, which ideally possesses the following characteristics:

$$R_2(\boldsymbol{x}, A) = CR_1(\boldsymbol{x}, A), \quad C > 0, \quad \sup_{\boldsymbol{x}, A} R_2(\boldsymbol{x}, A) \leq 1 \tag{42}$$

If a constant $C$, typically represented as $\frac{1}{\sup_{\boldsymbol{x}, A} R_1(\boldsymbol{x}, A)}$, can be identified, the following equation holds:

$$\begin{aligned} Z_t =& R_2(\boldsymbol{x_t}, A_t) + \gamma R_2(\boldsymbol{x_{t-1}}, A_{t-1}) + \gamma^2 R_2(\boldsymbol{x_{t-2}}, A_{t-2}) + \cdots \\ \leq& 1 + \gamma 1 + \gamma^2 1 + \cdots \leq \frac{1}{1 - \gamma} \end{aligned} \tag{43}$$

Taking into account the upper bound mentioned above, we recommend configuring the hyperparameters $V_{MAX} = \frac{1}{1-\gamma}$.

### 6.5.2 LOGARITHMIC OPERATIONS

If logarithmic operations are directly used to compute the loss function, it will result in significant precision loss, especially when dealing with very small values. Therefore, directly using the logarithm operator is unwise; we need to make some transformations on paper to avoid these precision losses. The technique demonstrated below is the 'log sum exp' trick.

$$\log(\sum_{1 \leq i \leq N} e^{x_i}) = x^* + \log(\sum_{1 \leq i \leq N} e^{x_i - x^*}), \quad x^* = \max_{1 \leq i \leq N} x_i \tag{44}$$

The above transformation ensures that the values inside the logarithmic operations are greater than 1, thereby avoiding the problem of significant precision loss when the values are very small. Based on the above discussion, 'log softmax' can be represented as:

$$\log(\frac{e^{x_j}}{\sum_{1 \leq i \leq N} e^{x_i}}) = x_j - x^* - \log(\sum_{1 \leq i \leq N} e^{x_i - x^*}), \quad x^* = \max_{1 \leq i \leq N} x_i \tag{45}$$

Furthermore, for the logarithmic operation of cumulative distribution, it can be represented as:

$$\log(1 - \frac{\sum_{1 \le i \le K} e^{x_i}}{\sum_{1 \le i \le N} e^{x_i}}) = \log(\frac{\sum_{K < i \le N} e^{x_i}}{\sum_{1 \le i \le N} e^{x_i}})$$

$$= (x^{**} + \log(\sum_{K < i \le N} e^{x_i - x^{**}})) - (x^* + \log(\sum_{1 \le i \le N} e^{x_i - x^*})) \tag{46}$$

$$x^* = \max_{1 \le i \le N} x_i, \quad x^{**} = \max_{K < i \le N} x_i$$

In practice, we found that setting a near-zero lower bound (such as $\epsilon = 0.0001$) for all cumulative probabilities when constructing the Policy Loss will be more robust. This helps prevent the actor network from making significant policy changes in pursuit of minor fluctuations in noise.

$$\log(1 - (1 - \epsilon)\frac{\sum_{1 \le i \le K} e^{x_i}}{\sum_{1 \le i \le N} e^{x_i}})$$

$$= \log(\frac{\epsilon \sum_{1 \le i \le K} e^{x_i} + \sum_{K < i \le N} e^{x_i}}{\sum_{1 \le i \le N} e^{x_i}})$$

$$= \log(\frac{\sum_{1 \le i \le K} e^{x_i + \log(\epsilon)} + \sum_{K < i \le N} e^{x_i}}{\sum_{1 \le i \le N} e^{x_i}}) \tag{47}$$

$$= x^{**} + \log(\sum_{1 \le i \le K} e^{x_i + \log(\epsilon) - x^{**}} + \sum_{K < i \le N} e^{x_i - x^{**}}) - (x^* + \log(\sum_{1 \le i \le N} e^{x_i - x^*}))$$

$$x^* = \max_{1 \le i \le N} x_i, \quad x^{**} = \max(\max_{1 \le i \le K} x_i + \log(\epsilon), \max_{K < i \le N} x_i)$$

