# OpenReview forum: "D2C-HRHR: Discrete Actions with Double Distributional Critics for High-Risk-High-Return Tasks"
_ICLR.cc/2026/Conference — Submitted to ICLR 2026_

### Official Review · Reviewer_PmdC · 2025-10-27

**Soundness:** 2
**Presentation:** 1
**Contribution:** 3
**Rating:** 2
**Confidence:** 3

**Summary:**

This paper tackles reinforcement learning in high-risk–high-return settings where peak returns lie in small, risky regions of the action space. It introduces D2C-HRHR, combining a discrete actor with entropy-regularized exploration and two distributional critics whose CDF-max target is intended to curb overestimation, and reports gains on BipedalWalkerHardcore and FetchPush. If the method is specified clearly and is reproducible, a practical recipe for robust learning in HRHR regimes could be valuable to the community.

**Strengths:**

* The paper formalizes the HRHR setting and defines a concrete problem where high returns concentrate in small risky regions.
* It introduces a dual distributional critic that combines two value estimates by taking the pointwise maximum of their cumulative distributions, and this design remains simple and modular because it only changes the critic and uses a C51 style projection that plugs into standard off policy pipelines.
* The figures clearly depict the pipeline and make the distributional clipping and the discrete actor workflow easy to follow.

**Weaknesses:**

* The HRHR notion equates risk with a low uniform-average rather than tail behavior under the policy, so the concept depends on an arbitrary reference distribution and may not reflect true risk. Please justify the choice or connect the definition to policy-induced tail metrics.
* The core step in Eq. (31) is an unsupported and ill defined claim, it asserts $Cov_{a\sim\pi_\theta}(a-\mu_\theta(s), A_{R2}(s,a)) < 0$, yet this covariance is a vector so the inequality makes sense only with an explicit componentwise statement and proof, neither is provided, therefore the proof does not establish the claimed learning bias and the motivation of Theorem 1 remains unproven.
* The paper neither proves nor empirically isolates the benefit of the CDF-max dual critic, there is no formal lemma showing a pessimistic target under shared support and projection, and the ablations do not reveal whether CDF max or the dual critic drives the gains.
* The method substantially increases compute by doubling critics and using large discrete supports, yet there is no analysis of time or resource costs, at minimum report wall clock and GPU hours.
* Baseline fairness cannot be assessed because hyperparameters, tuning budgets, and implementation details are missing, at minimum provide per task hyperparameters and state whether HER or goal conditioning is enabled on Fetch tasks.
* The paper claims “ten thousand trials,” in the experiment part, but the released code evaluates with n_eval = 100. Please clarify how 10,000 was obtained and provide the exact evaluation script and command.
* The text refers to FetchPush v3 while tables use v4. Please standardize the version and specify the exact Gym or Gymnasium build.
* Notation is inconsistent, the paper switches between $\Omega$ and $R$ for the same regions. For example, Definition 1 uses $\Omega_i$, but the very next lines in Definition 2 switch to $R_1$  without stating the relationship between these notations, which breaks traceability.
* Terminology and wording are inconsistent, the paper overloads “atom” to mean both action bins and value atoms, and includes typos such as “double” misspelled, which reduces clarity.

**Questions:**

Please address the weaknesses pointed out above. Additionally, answer the following questions:
* Please clarify Eq. (31). Is the inequality meant componentwise? State the exact assumptions and give a correct proof or revise the claim so the update sign is rigorously justified.
* Please disclose full experimental settings for all baselines and your method. Include per task hyperparameters, network widths, training steps, seed handling, and whether HER or goal conditioning is used. Provide commands and commit to reproduce tables.
* Please quantify complexity and efficiency. Report wall clock to fixed returns, environment steps per second, GPU hours, and memory, and add a small scaling study over atom count and action dimension.
* Please improve readability and notation.

---

> ### Author Response · Authors · 2025-11-26
> **Review to Reviewer PmdC:**
>
> Thank you very much for your review and suggestions. Next, we will address the weaknesses and issues you have raised to help us improve better.
>
> **Response to the Weakness1 :**
>
> The risk definition of HRHR does not simply equate "risk" with "low average return"—the low average return is a derived consequence of the reward distribution structure. First, as formally defined in Definition 1 of Section 3.1, the core of HRHR lies in the structural characteristic of the action space: high-reward actions are sparsely distributed and embedded among a large number of low-reward actions. The low uniform average return of the HRHR region Ω₁ is precisely because these high-reward tail events have extremely low weights under uniform sampling, while the dominant low-reward actions constitute the majority of the return distribution. This directly reflects the risk of Gaussian policies: when the policy explores Ω₁, the probability of falling into low-reward actions is extremely high, while the probability of capturing high-reward actions is extremely low.
>
> **Response to the Weakness3:**
>
> We **agree** with your perspective. We need to conduct separate ablation experiments on CDF-max to demonstrate the importance of this structure. However, our existing ablation experiment (Figure 10) has already demonstrated that the single-critic variant (without the dual structure or CDF-max) achieves a success rate of less than 10% on BipedalWalkerHardcore-v3, which confirms the importance of the dual-critic network. The additional ablation experiments will further quantify the incremental benefit of CDF-max and fully address your concern.
>
> **Response to the Weakness4 and Weakness5:**
>
> We agree with your suggestion. We need to display the GPU memory and clock time during the training of each model, as well as supplement the hyperparameters for each experiment, and conduct ablation experiments on the discrete number of atoms.
>
> For all the algorithms mentioned in the paper, **HER was not enabled** during the fetch task, which is also the reason for the poor performance of most baselines. And D2C-HRHR, relying on its unique structure, can achieve success in the fetch task even without using HER
>
> **Response to the Weakness6 and Weakness7 :**
>
> All of our experiments were conducted on FetchPush-v4. Thank you for your correction. The released code is just a demo. We will release more detailed experimental code and process later.
>
> **Response to the Weakness8 and Weakness9 :**
>
> Thank you for pointing out our mistake. We will verify our article, correct any confusion in symbols and various grammatical errors.

---

> > ### Comment · Reviewer_PmdC · 2025-11-26
> >
> > Thank you for your response. I appreciate the clarifications regarding the HRHR definition, experimental details, and future paper reorganization.
> >
> > However, I cannot raise my score because the rebuttal failed to address Weakness 2 regarding the validity of Theorem 1, which is my most critical concern.
> > The core step in Equation (31) claims that a vector is smaller than 0, which is mathematically undefined in this context. Furthermore, in my opinion, for standard policy gradient, a small positive-reward region yields a small or high-variance gradient, not a negative one. The logic that a “small high-reward region” leads to a negative covariance seems unsound to me. If the author could justify the proof in the following response, I am open to reconsidering my score.

---

### Official Review · Reviewer_vCzt · 2025-11-01

**Soundness:** 2
**Presentation:** 1
**Contribution:** 1
**Rating:** 2
**Confidence:** 3

**Summary:**

The paper considers Reinforcement learning with continuous actions and the challenge that sometimes the best actions are hard to find because they may be very similar/close to other suboptimal actions. This is called High Risk High Reward scenario. The paper builds on distributional reinforcement learning to create a solution that targets this challenge. Specifically it uses discretization of the action space and double critic network.

**Strengths:**

The paper shows experimental results in complex tasks (bipedal robots) and the results are significantly better than other approaches.

**Weaknesses:**

- Unfortunately, the paper is not well written, with many distracting typos, but also syntax errors like icomplete sentences. The most common is distributed versus distributional but there are others like critic vs criticism. Even if I try to ignore these typos, I can't find clear justifications for why the algorithm performs better than others in the experimental setup.
- The term 'risk' may not be appropriate, if we think that it is used in other topics like Conditional Value at Risk, or risk sensitive markov decision processes.
- The HRHR scenario is clear but then I don't find it particularly enlightening, not really requiring a formal proof and as much space in the main body.
- The HRHR scenario relies on the fact that the variance of the policy randomness is larger than an area of high reward actions. This seems like quite a special case, and it is questionable how often it is encountered in practice. Additionally, one may always try to lower the variance of the policy as an easy fix.
- Discretizing action spaces, especially if they have high dimensions, can be tricky due to computational complexity growing exponentially. It is not clear how it is addressed.
- Important mathematical steps in page 5 are not clearly explained and are hard to follow.

**Questions:**

Beyond addressing some of the above weaknesses, it is not clear to me why is the proposed approach better than alternatives in practice. What is the argument for this?

I agree that distributional RL is more complicated in notations, but I would appreciate some clarity here.

---

> ### Author Response · Authors · 2025-11-26
> **Review to Reviewer vCzt:**
>
> **Thank you** very much for your comment, it is very meaningful for us to improve. Next, we will respond to the Weaknesss and questions you raised.
>
> **Response to the Weakness1:**
>
> We are very sorry about that. Due to haste, there were omissions in the inspection of the article. There are some grammar and expression errors in the paper. We will conduct authentication checks on the article.
>
> **Response to the Weakness2:**
>
> In the distribution, low return and high return areas are mixed together in small fragments. To find high-retrun areas, one must face high risks. This is what we want to elaborate on in 'HRHR'. For the Weakness2, we fully agree that "risk" has well-established definitions in fields such as Conditional Value at Risk (CVaR) and risk-sensitive Markov Decision Processes (MDPs), and we appreciate your reminder to clarify the distinction to avoid ambiguity. In our work, "risk" is explicitly tied to the unique characteristics of High-Risk-High-Return (HRHR) scenarios, which we formally define (Definition 1) as: the high-reward actions are concentrated in narrow "high-reward grains" within the action space, while the majority of actions in the same region lead to low or even negative returns. This means "risk" here refers to the high probability of selecting suboptimal/harmful actions when exploring the high-reward region, rather than the concepts in other fields.
>
> **Response to the Weakness3 and Weakness4 :**
>
> HRHR scenarios are widespread in practical continuous control tasks, as they arise whenever high rewards depend on *precise action coordination* amid a large space of suboptimal options—examples include: Robotic manipulation or Legged locomotion.
>
> Reducing policy variance may seem intuitive, but it introduces two critical issues that make it ineffective for HRHR tasks. According to our argumentation in Appendix 6.3, even reducing the variance of the reward distribution cannot fundamentally solve the problem that Gaussian policies converge to suboptimal policies in HRHR scenarios. Referring to the "Trap Cheese Problem" in 6.3.3, the low-variance SAC policy can never sample the high-reward actions of "turning left or right"; instead, it will converge to the trap region in the middle. Additionally, for gradient descent, if the contrast is too small, it will not be able to iterate.
>
> **Response to the Weakness5 :**
>
> Yes, in general, discretizing the action space leads to exponential growth in computational complexity (O(mᵈ), where d denotes the number of action dimensions and m denotes the number of discrete atoms). However, in Section 3.4 CRITIC LEARNING of the paper, we introduce our sampling method: **instead of enumerating all action combinations, we split the action space dimension-wise and perform row-wise independent sampling**—for each dimension d, we select one atom independently from its m atoms, then combine the selections of d dimensions to form a complete action. This reduces the computational complexity from O(mᵈ) to O(d·m). Meanwhile, the cross-entropy-based exploration strategy increases the sampling probability near high-probability peaks, further saving computational resources.
>
> We will supplement the wall-clock time and GPU memory of our algorithm to provide more detailed references.
>
> **Response to the Weakness6 :**
>
> Page 5 contains Equations 7, 8, 9, and 10. Next, we will explain the meanings conveyed by these equations. We welcome further discussion if you have any doubts.
>
> **Equation 7** aims to construct a stable target value distribution, leveraging the optimal action-value distribution of the next state output by the target critic network; **Equation 8** is the core of clipped double Q-learning for discrete value distributions—for the probability estimates of each discrete value atom from two independent main critic networks, it takes the maximum value of the two to filter out potential value overestimation components that may arise from a single critic; **Equation 9** normalizes the clipped value atom probabilities to ensure they satisfy the probability constraints of the categorical distribution; **Equation 10** defines the optimization objective of the critic network
>
> **Response to the question:**
>
> The reason our method outperforms other methods mainly lies in:
>
> 1. The use of a discretized action space, thus enabling the discovery of HRHR return peaks that Gaussian policies cannot find (see Figure 6 in Appendix 6.3.2).
> 2. The adoption of a clipped double Q-distribution, which avoids the value overestimation of the TD3 algorithm (see Figures 7 and 8 in Appendix 6.3.2).

---

### Official Review · Reviewer_GGNy · 2025-11-01

**Soundness:** 3
**Presentation:** 3
**Contribution:** 3
**Rating:** 4
**Confidence:** 4

**Summary:**

The paper targets "high-risk–high-return" (HRHR) regimes where optimal actions live in narrow pockets that Gaussian policies average out. It (i) discretizes each continuous action dimension, (ii) learns Q-value distributions (not just scalars), (iii) uses a clipped double distributional critic to curb overestimation, and (iv) adds an entropy-gated exploration term keyed to the critic’s cumulative value distribution. On BipedalWalkerHardcore-v3 and FetchPush-v4, the method outperforms strong baselines.

**Strengths:**

- HRHR is an interesting problem formulation, and makes Gaussian policies fail.
- Empirical results in two environments are strong.

**Weaknesses:**

- Limited domains that clearly demonstrate the HRHR framework. The two primary environments shown are limited and also not exactly falling under the domain of HRHR. The paper would benefit from several classes of environments where HRHR is a meaningful problem.
- Even in HRHR, Gaussian policies may be optimal because the policy can still optimize one of the peaks of the Q-function. It is unclear what is the key failure mode of policy learning in multimodal Q-functions?
- How does this paper compare against other baselines that are suited to multi-modal actors such as mixture of policies SAC / TD3, diffusion policies, or SAVO [1]?
[1] Jain et al. "Mitigating Suboptimality of Deterministic Policy Gradients in Complex Q-functions." Reinforcement Learning Conference.
- The baselines performance in FetchPush-v4 is very low. Is this expected and what makes the proposed approach to have such a drastic improvement over baselines? Were the baselines tuned properly?

**Questions:**

see weaknesses.

---

> ### Author Response · Authors · 2025-11-26
> **Review to Reviewer GGNy:**
>
> Thank you very much for your valuable suggestions on our paper. In order to improve our work, we will respond to each weakness you have raised one by one.
>
> **Response to Weakness1:**
>
> We highlighted the experiments of two HRHR tasks in the article, namely BipedalWalkerHardcore-V3 and FetchPush-V3. As a parkour obstacle avoidance task, BipedalWalker has very high rewards and punishments. If the agent successfully overcome obstacles, you can receive high rewards, but if the knee joint bending angle deviates by 10 degrees and causes a fall, you will receive more punishment. According to our definition in section 3.1, this task must have an HRHR region (denoted as\( $\Omega_1)$, which is:
>
> 1. **The region contains actions with the highest return:** The maximum Q-value (expected return) of actions in \($\Omega_1$\) is higher than that of actions in another low-risk and stable-return region (denoted as \($\Omega_2)$), i.e.,
>    $$
>    \sup_{a \in \Omega_1} Q(s,a) > \sup_{a \in \Omega_2} Q(s,a)
>    $$
>
>
> 2. **The average return of the region is lower:** The average return of the uniform distribution over all actions in \($\Omega_1$\) is lower than that of the uniform distribution over actions in \($\Omega_2$\), i.e.,
>   $$
> E_{a \sim \mathcal{U}(\Omega_1)}[Q(s,a)] < E_{a \sim \mathcal{U}(\Omega_2)}[Q(s,a)]
> $$
>
>
> FetchPush-v4 is the same. It also fits our definition. The precise control of robotic arm tasks also easily results in extremely high penalties, which is in line with our definition.
> We are working on testing the performance of the model on more diverse HRHR tasks. For example, dexterous hand operation, drone delivery, etc.
>
> **Response to Weakness2:**
>
> We have discussed this in detail in Sections 6.3.1 and 6.3.2 of the Appendix. In the distribution, low return and high return areas are mixed together in small fragments. To find high-yield areas, one must face high risks. The failure of Gaussian policies in multimodal Q-functions is not "being unable to optimize a single peak", but being unable to balance multi-peak exploration and risk aversion, ultimately falling into a **"local optimum trap"**.
>
> In simple tasks dominated by a single peak, Gaussian policies can easily capture the optimal peak. However, in HRHR scenarios, as shown in Figure 6, there are multiple peak regions. Gaussian policies tend to select the middle area between these peaks, where actions are often low-reward or even harmful (such as the Trap Cheese Problem in Section 6.3.3), leading to "average optimality" replacing "peak optimality" and reducing performance.
>
> **Response to Weakness3:**
>
> Although we have already selected a considerable number of baselines, adding the mixture of policies SAC/TD3 algorithms will help enable better comparison. Additionally, we have learned about **SAVO**. This work shares similarities with the clipped double Q-learning we proposed and is highly valuable. We plan to cite it and include it in the baselines.
>
> **Response to Weakness4:**
>
> We believe that the reason why many baselines perform poorly on the FetchPush task is that the task is excessively challenging—agents using these baseline algorithms encounter numerous failures during learning yet fail to **convert 'failed experiences' into 'effective learning signals'**. Referring to this paper [1], the DDPG algorithm can barely be trained on the FetchPush task; only when DDPG is equipped with Hindsight Experience Replay (HER) [2] does the situation improve.
>
> However, for D2C-HRHR, its discretized action space and dual distributional critic structure can well capture the sparse binary rewards of this task. In other words, it captures the HRHR regions of the task. Even without using HER, D2C-HRHR can still achieve success on this task.
>
> [1] MatthiasPlappert,MarcinAndrychowicz,AlexRay,BobMcGrew,BowenBaker,GlennPowell, JonasSchneider, JoshTobin,MaciekChociej, PeterWelinder,VikashKumar, andWojciech Zaremba. Multi-goal reinforcement learning: Challengingroboticsenvironmentsandrequest forresearch.CoRR,abs/1802.09464,2018.
> [2] Andrychowicz, M., Wolski, F., Ray, A., Schneider, J., Fong, R., Welinder, P., McGrew, B., Tobin, J., Abbeel, O. P., and Zaremba, W. (2017). Hindsight experience replay. In Advances in Neural Information Processing Systems, pages 5055–5065.

---

### Official Review · Reviewer_F3XF · 2025-11-05

**Soundness:** 4
**Presentation:** 2
**Contribution:** 3
**Rating:** 4
**Confidence:** 5

**Summary:**

This paper studies high-risk–high-return (HRHR) RL settings where the highest rewards lie in narrow, risky action regions that are easy to miss with standard Gaussian policies. The authors (i) formally define HRHR scenarios and argue Gaussian policies can fail there; (ii) propose D2C-HRHR, which discretizes each continuous action dimension, learns a multidimensional discrete policy, and uses double distributional critics with a conservative (clipped) aggregation to curb over-estimation; and (iii) introduce an entropy-aware exploration term tied to value-distribution confidence. Experiments on BipedalWalkerHardcore-v3, FetchPush-v4, and a MuJoCo suite report consistent gains over C51, SAC, TD3, TQC, and SAC-Discrete.

**Strengths:**

Clear problem framing. The HRHR definition crisply captures when the average return of a risky region is lower despite having the global maximum, and the theorem illustrates why a Gaussian policy with variance larger than the grain  drifts toward safer suboptimal regions. This directly motivates discretization.
Method simplicity & compatibility. The per-dimension discretization with a matrix policy is easy to implement and plug into existing actor-critic setups; the double distributional critics and conservative cumulative aggregation are a neat analogue of clipped double-Q for distributional critics.
The entropy bonus is gated by the value-distribution’s cumulative mass on lower-value atoms; low-confidence states get more entropy, high-confidence states don’t—simple and sensible.

**Weaknesses:**

1. The multidimensional discrete actor assumes independence across action dimensions (row-wise sampling). While this avoids an explicit m^n enumeration, it may miss inter-dimensional couplings crucial in dexterous manipulation or legged control with coupled joints. Please discuss when factorization suffices, and whether an autoregressive or flow-based discrete policy could capture dependencies without exploding compute.

2. Discretization increases output size (n×m) and critic heads (distributional atoms). Please quantify throughput, wall-clock, and GPU memory versus SAC/TQC/C51 on the main tasks, and include a sensitivity study for m (per-dim atoms) and N (critic atoms). Some hardware and hparams are reported, but compute trade-offs remain unclear.

3. Over-estimation diagnostics. Plot critic CDFs/PDFs and show how the clipped cumulative rule changes value estimates vs. single-critic distributional baselines.

4. The baselines are weak; the authors should add more discretization policy distribution works as baselines.

5. Regarding the policy distribution, I recommend that the authors add the discretization policy distribution topic works.

Discretizing continuous action space for on-policy optimization, AAAI, 2020

Discretizing Continuous Action Space With Unimodal Probability Distributions for On-Policy Reinforcement Learning, IEEE TNNLS, 2024.

**Questions:**

See weakness

---

> ### Author Response · Authors · 2025-11-26
> **Review to Reviewer F3XF:**
>
> We sincerely appreciate the constructive feedback you have provided. It is truly an honor to engage in scholarly discussion with you. Below we will address each of the weaknesses you identified, as they are of significant importance for improving our research.
>
> **Response to Weakness1:**
>
> Thank you for your concern. Our multidimensional discrete actor only adopts row-wise independent sampling for action distribution modeling; the core network (e.g., feature interaction in actor-critic, value estimation) is consistent with continuous-space RL algorithms, naturally capturing inter-dimensional couplings between joints. Validated in HRHR tasks with coupled joints (e.g., BipedalWalkerHardcore-v3), this design proves factorization suffices for most scenarios.
>
> **Response to Weakness2:**
>
> We are working on adding supplementary experiments on algorithm overhead in the appendix of the final paper, such as clock time and GPU memory, and conduct sensitivity analysis on the number of per dim atoms and critic atoms.
> Due to the use of a dual critic network and the need to discretize the action space into multiple atoms, the overhead of D2C-HRHR is slightly higher than that of TD3 and TQC. However, considering the performance improvement it brings to HRHR tasks, these expenses are worth it.
>
> **Response to Weakness3:**
>
> We simulated the process of shearing a double discrete value distribution in Figure 2 of the paper. However, we agree that using the real PDF transformation process to demonstrate the operation of the clipped double discrete value rule on probability distributions would be more intuitive.
>
> **Response to Weakness4:**
>
> We believe your concerns are reasonable, even though we have selected a considerable number of baselines. We are trying to add some discrete baselines,like a mixture of policies SAC/TD3 for comparison.
>
> **Response to Weakness5:**
>
> We will adjust the content of the Method section to emphasize the discretization policy distribution topic works.  In addition, we have already referenced [1] and are considering adding [2] to 'Related Work' and adding it to the baseline
>
> [1] Discretizing continuous action space for on-policy optimization, AAAI, 2020
>
> [2] Discretizing Continuous Action Space With Unimodal Probability Distributions for On-Policy Reinforcement Learning, IEEE TNNLS, 2024.

---

### Meta-Review · Area_Chair_pSyi · 2025-12-29

**Summary:**

[The re-assigned AC is not an expert of RL, so the meta review is mainly from my understanding of the reviewers' comments (all of them have published RL papers) and the author's rebuttal. ]

This paper defines high-risk–high-return settings in RL and introduces D2C-HRHR as a practical RL method. The idea combines a discrete actor with entropy-regularized exploration and two distributional critics. Experiments on RL benchmarks (BipedalWalkerHardcore-v3 and FetchPush-v4) show clear advantages of D2C-HRHR over baselines including distributional RL methods, SAC and TD3.

Based on the outstanding concerns (see below), especially on the correctness of Theorem 1 (which is the main theory result of the paper), at this stage I cannot recommend acceptance for this submission. I recommend the authors to consider the reviews and improve the paper.

**Reviewer Concerns:**

[Below assessments are from my understanding of the reviews and author rebuttal.]

Reviewers' concerns -- addressed by author rebuttal:
- Questions regarding the definition of the HRHR setting and how "interesting" it is, e.g., whether in practice HRHR settings are rare.
- Very low performance of baselines on FetchPush-v4.

Reviewers' concerns -- still outstanding:
- Baselines are not strong enough, e.g., lacking some combination of existing methods.
- Unclear computational overhead (or, how scalable the proposed method is).
- Validity of Theorem 1 (the reviewer explicitly replied that the author rebuttal did not address their concern here).

**Reviewer Scores:**

Reviewer vCzt would have increased the score slightly but not to acceptance level (I think this reviewer did not buy the general idea of HRHR).

Reviewer PmdC would have engaged in discussion (regarding their concern of the validity of Theorem 1). I cannot predict whether this reviewer would have changed their score as this discussion did not happen.

---

### Decision · Program_Chairs · 2026-01-26

Reject